# Observational operator for fair model evaluation with ground NO$_2$ measurements

Li Fang[1], Jianbing Jin*[1], Arjo Segers[2], Ke Li[1], Ji Xia[1], Wei Han[3], Baojie Li[1], Hai Xiang Lin[4,5], Lei Zhu[6,7,8], Song Liu[6], and Hong Liao*[1]

[1]Joint International Research Laboratory of Climate and Environment Change, Jiangsu Key Laboratory of Atmospheric Environment Monitoring and Pollution Control, Jiangsu Collaborative Innovation Center of Atmospheric Environment and Equipment Technology, School of Environmental Science and Engineering, Nanjing University of Information Science and Technology, Nanjing, Jiangsu, China
[2]TNO, Department of Climate, Air and Sustainability, Utrecht, The Netherlands
[3]CMA Earth System Modeling and Prediction Centre, Chinese Meteorological Administration, Beijing, China
[4]Institute of Environmental Sciences, Leiden University, Leiden, The Netherlands
[5]Delft Institute of Applied Mathematics, Delft University of Technology, Delft, the Netherlands
[6]School of Environmental Science and Engineering, Southern University of Science and Technology, Shenzhen, China
[7]Guangdong Provincial Observation and Research Station for Coastal Atmosphere and Climate of the Greater Bay Area, Shenzhen, China
[8]Shenzhen Key Laboratory of Precision Measurement and Early Warning Technology for Urban Environmental Health Risks, School of Environmental Science and Engineering, Southern University of Science and Technology, Shenzhen, China

**Correspondence:** Jianbing Jin (jianbing.jin@nuist.edu.cn) and Hong Liao (hongliao@nuist.edu.cn)

**Abstract.** Measurements collected from ground monitoring stations have gained popularity as a valuable data source for evaluating numerical models and correcting model errors through data assimilation. The penalty quantified by simulation-minus-observations drive both model evaluation and assimilation. However, the penal forces are challenged by the existence of a spatial scale disparity between model simulations and observations. Chemical Transport Models (CTMs) divide the atmosphere into grid cells, providing a structured way to simulate atmospheric processes. However, their spatial resolution often does not match the limited coverage of in-situ measurements, especially for short-lived air pollutants. Within a broad grid cell, air pollutant concentrations can exhibit significant heterogeneity due to their rapid generation and dissipation. Ground observations with traditional methods (including nearest search and grid mean) are less representative when compared to model simulations. This study develops a new land-use-based representative (LUBR) observational operator to generate spatially representative gridded observation for model evaluation. It incorporates high-resolution urban-rural land use data to address intra-grid variability. The LUBR operator has been validated to consistently provide insights that align with satellite OMI measurements. It is an effective solution to accurately quantify these spatial scale mismatches and further resolve them via assimilation. Model evaluations with 2015-2017 NO$_2$ measurement in the study area demonstrate biases and errors differed substantially when the LUBR and other operators were used, respectively. The results highlight the importance of considering fine-scale urban-rural differences when comparing models and observations, especially for short-lived pollutants like NO$_2$.

# 1 Introduction

Air pollution is acknowledged as a significant risk factor for chronic non-communicable diseases for its contribution to global morbidity and mortality, surpassing all other known environmental risk factors (Al-Kindi et al., 2020). Despite considerable improvements in air quality in recent years globally, many regions still suffer from severe air pollution, impacting the living conditions of their residents (Li et al., 2021). Numerical models are fundamental tools in modern science, used across disciplines to describe complex systems, analyze observations, test hypotheses, and project future behavior. They are pivotal in atmospheric science, serving as central tools for weather prediction, climate research, and extensively describing atmospheric dynamics (Brasseur and Jacob, 2017). Atmospheric chemistry transport models (CTMs) utilize mathematical equations to represent the intricate relationships between atmospheric concentrations of chemical species and the factors influencing them, such as emissions, transport, chemistry, and deposition processes. These models can simulate the temporal-spatial patterns of air pollutants from the past to the future, aiding policymakers in identifying the most effective strategies for reducing emissions (Liu et al., 2018; Zhai et al., 2021; Jin et al., 2023).

The rapid advance in computing power and atmospheric science has facilitated the development and widespread use of numerous three-dimensional CTMs, such as GEOS-Chem (Bey et al., 2001), CESM2 (Danabasoglu et al., 2020), WRF-Chem (Grell et al., 2005), etc., over the past few decades. Undoubtedly, these models serve as powerful tools to investigate and simulate the intricate behavior of atmospheric composition and chemical processes. However, these models cannot perfectly reproduce the true atmospheric dynamics due to various factors. Matthias et al. (2018) has highlighted persistent uncertainties in input data, including emission inventories and meteorological data. The model parameterization and simplifications are also not perfect (Stensrud, 2009), and addressing knowledge gaps in chemical reaction mechanisms remains a challenge. Moreover, CTMs face difficulties in accurately representing atmospheric processes at fine spatial scales and capturing rapid temporal variations (Goodkind et al., 2019). This challenge stems primarily from the high computational demands of conducting high-resolution or long-term simulations (Bindle et al., 2021).

Observations, unlike CTMs simulations, offer a measurement of the real-world environment by utilizing a range of instruments, sensors, and techniques. Ground observation data is widely regarded as the most fundamental measurement, and usually serves as a benchmark for calibrating the accuracy of other data, such as model results (Fang et al., 2022) and satellite data (Garane et al., 2019). Since 2013, the China Ministry of Environmental Protection (MEP) has established over 1800 ground-based stations dedicated to measuring primary pollutants including $PM_{2.5}$, $PM_{10}$, $NO_2$, $SO_2$, CO and $O_3$ (Sheng and Tang, 2016). These ground observations provide valuable insights into air pollution conditions and are widely used for model evaluations (Zhu et al., 2021), and their distributions are presented in Supplement Figure S2. Concurrently, the rapid advancements in satellite remote sensing and other technologies have made it possible to observe near-surface air pollutant abundances from space (Xu et al., 2019; Kim et al., 2021). For example, satellite onboard instruments such as the Ozone Monitoring Instrument (OMI) and the Tropospheric Monitoring Instrument (TROPOMI) can facilitate the measurement of $NO_2$ with extensive coverage (van Geffen et al., 2022). This study primarily focuses on analyzing the disparities between model simulations and observations of $NO_2$ and fine particulate matter ($PM_{2.5}$).

Measurements collected by ground monitoring stations and satellite instruments are widely used for model evaluations and for correcting model errors through the application of data assimilation techniques (Kalnay, 2002). Mathematically, observations and simulations with different scales and dimensions are not comparable directly in the model evaluations. Observations from satellites typically have finer spatial resolution than model simulations, so the comparison between them is less affected by spatial scale disparity. Conversely, ground observations are sparse and uneven, making it more challenging to compare them with the gridded simulations. To address this, two prevalent pre-processing methods are often employed. The first one entails calculating the average value of all the observations located in a given model grid (Dang and Liao, 2019; Dai et al., 2023), then compared to the gridded simulation. The second method conducts the nearest search for model values corresponding to any given measurements (Jin et al., 2021). A third approach could be only using monitoring stations that are spatially and temporally representative for the model grid cells. However, there is no standard definition for determining the extent to which monitoring stations can represent model grids. Additionally, this method may result in the unavoidable loss of valuable ground observations. In the subsequent sections of this paper, these two methods will be illustrated in detail, and they are referred to as 'grid mean' and 'nearest search', respectively. With this, the observation-minus-simulation discrepancy can be calculated and serves as the driving force in determining the extent to how much the uncertain model parameters or states are adjusted during the evaluation or assimilation process. When observation biases are present together with the model errors, there is a danger of misleading model evaluation or divergent model estimation in the assimilation (Lorente-Plazas and Hacker, 2017). This is because failing to account for these biases properly can lead to inaccurate attribution of the error sources. Previous studies (Bédard et al., 2015; Eyre, 2016; Jin et al., 2019) have highlighted the significance of addressing observation biases and their correction.

The existence of a spatial scale disparity between model simulations and observations is a persistent challenge (Schutgens et al., 2016). The aforementioned two commonly used methods for model evaluation can potentially cause large representative errors of observations. The CTMs divide the atmosphere into a series of horizontal and vertical grid cells. Each grid cell represents the mean state in a specific region (Yan et al., 2016). As an example, for GEOS-Chem, the nested simulation typically adopts a relatively high horizontal resolution of 0.5°latitude by 0.625°longitude, which is widely used in practice keeping the balance between the complexity and computing power (Wang et al., 2004; Chen et al., 2009; Wang et al., 2013; Yan et al., 2016). However, in-situ measurements are typically limited to a few kilometers of the surrounding atmosphere (Pattinson et al., 2014; Schutgens et al., 2016), and the effective spatial range for short-lived gases is even more restricted. For instance, concentrations of ground $NO_2$ (with a lifetime of approximately several hours as noted in Shah et al. (2020b)) exhibit significant variations between urban and rural areas (Pattinson et al., 2014). This discrepancy arises due to anthropogenic $NO_2$ emissions primarily occurring in the troposphere, stemming from sources such as transportation, industrial production, and power plants (Li et al., 2017). The concentration of $NO_2$ diminishes considerably as the distance from the emission source increases, owing to its rapid consumption through the process of photolysis after its production (Finlayson-Pitts and Pitts Jr, 1999). Consequently, the distribution of $NO_2$ concentrations within a large grid cell is highly heterogeneous, making it challenging to accurately represent the true average concentration of the grid solely by directly using the values of several monitoring stations within the grid or simply averaging them. Meanwhile, as most of the ground monitoring sites such as the

China MEP network are located in the severe-polluted urban areas, this further prevents them from fairly representing the mean status of the actual atmospheric environment.

In this study, we proposed a land use-based representative (LUBR) observational operator to represent real atmospheric pollutant concentrations, using both the ground observations and the land use information. The land use information is acquired from nighttime light (NTL) data which can distinguish between urban and rural areas. This new operator was compared alongside two other commonly used observational operators ('grid mean' and 'nearest search') to evaluate their performance for model evaluation. Our novel observational operator was applied in both $NO_2$ and $PM_{2.5}$ model evaluations. The latter has a relatively longer atmospheric lifetime of several days compared to the former (several hours to one day). The temporal scope of this study spans from 2015 to 2017. Overall, the LUBR method incorporates high-resolution land use data to account for intra-grid variability and generate observation datasets that are more spatially representative. This helps address the scale mismatch between models and observations that have impaired robust evaluation, especially for short-lived gases like $NO_2$.

This study is structured as follows. Section 2.1 and Section 2.2 describe the study domain, observations, and model used. Details on the urban/rural factors and the LUBR algorithm are provided in Section 2.3 and Section 2.4. Section 3.1 first provides the model validation, followed by the revelation of discrepancies between observations and model simulations in Section 3.2. The comprehensive evaluation of the LUBR operator is then presented in Section 3.3. Next, Section 3.4 discusses the spatial and temporal evaluations of $NO_2$ and $PM_{2.5}$ pollutants either using LUBR or using the traditional grid mean/nearest search methods. Statistical metrics quantifying their performance are also analyzed. Finally, the key findings and implications of developing such a spatially representative observational operator are summarized in the conclusion.

## 2 Data and methods

This chapter begins by introducing the study domain and observations in Section 2.1. Following that, we present the GEOS-Chem model utilized in our research in Section 2.2. Moving forward to Section 2.3, we explore the variations in air pollutant concentrations between urban and rural areas, along with an introduction to the dynamic factors associated with these areas. Lastly, Section 2.4 offers an in-depth description of the LUBR algorithm for building the observational operator.

### 2.1 Study domain and observations

This study investigates how our observational operator benefits air quality model evaluations over the whole study area as presented in the left panel of Fig. 1. To provide a more comprehensive insight, this study focuses on two regions characterized by severe $NO_2$ pollution: the North China Plain (NCP; 34–41° N, 113–119° E) and the Yangtze River Delta (YRD; 30–33° N, 119–122° E). These regions are examined in greater detail for a more elaborate illustration.

### 2.1.1 Ground observations

When assessing model simulations of ground-level $NO_2$ against in-situ ground observations in China, it is consistently observed that the model tends to underestimate these observations at most monitoring stations. A widely acknowledged ex-

planation for this phenomenon is that the environmental monitoring stations established by the China MEP are predominantly situated in urban areas (as shown in Figure S3). This geographical bias may contribute to an overestimation of grid-scale ground-level $NO_2$ observations across the study area. Panels a and b of Fig. 1 are partially enlarged views of regions with significant local urbanization in NCP and YRD. Grid lines represent the simulated grids of longitude and latitude in GEOS-Chem, where urban sites are marked with blue dots and rural sites with red squares. Three primary types of grid cells are present: *U* contain solely urban sites, *R* with only rural sites, *Mix* encompass both urban and rural monitoring stations, while the rest lack any sites altogether. It's noteworthy that the urban area percentage within a model grid, as derived from annual nighttime light data, significantly differs from the percentage of urban sites present within that same grid as shown in Fig. 1(a) and (b).

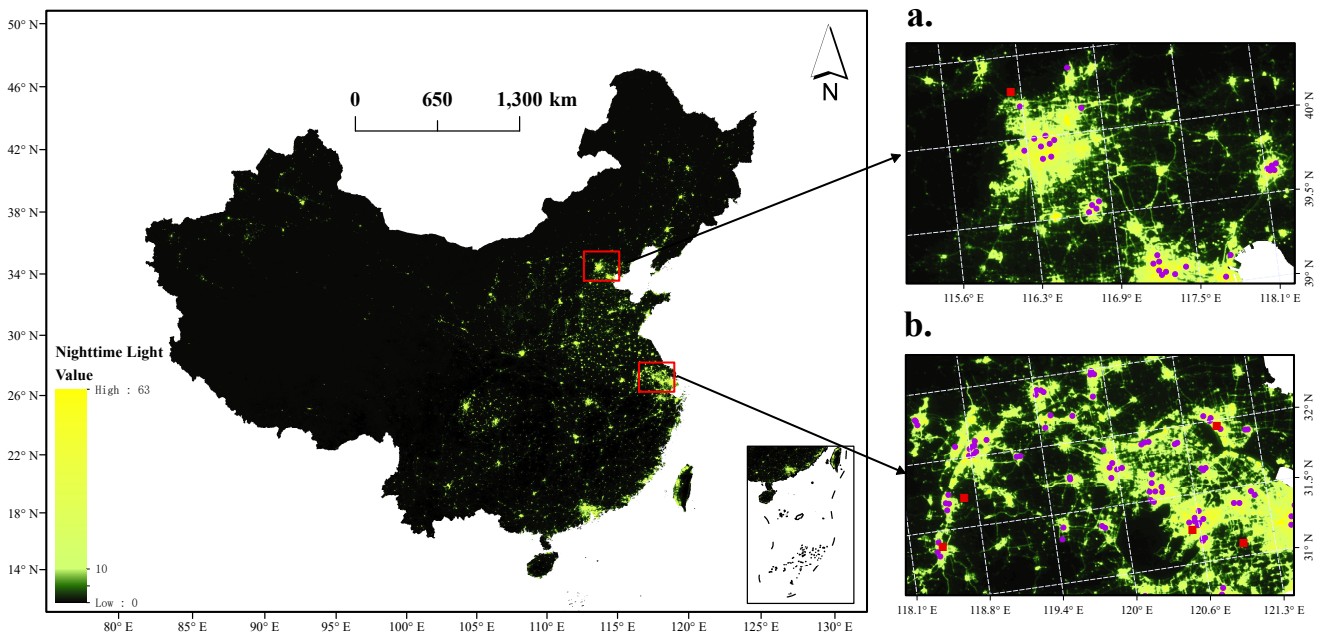

**Figure 1.** The left subplot shows the night lights in the study area derived from V2.1 annual global VIIRS nighttime lights, with data averaged for the year 2020. The intensity of color corresponds to the level of urbanization, where brighter colors indicate higher urbanization levels. Subplots a and b display regions with significant local urbanization in NCP and YRD, respectively. In these subplots, purple dots and red rectangles are used to represent urban monitoring stations and rural monitoring stations, respectively.

### 2.1.2  OMI/Aura observations

Launched aboard the NASA EOS Aura satellite on July 15, 2004, OMI operates within a sun-synchronous ascending polar orbit. OMI conducts simultaneous measurements across a swath spanning 2600 km, partitioned into 60 Fields of View (FOVs). These FOVs range in dimension from approximately 13km x 24km near Nadir to around 24km x 160km at the outermost FOVs. OMI provides observations only around 13:45(local time) overpassing window and is most reliable under clear-sky conditions. The $NO_2$ total column concentrations utilized in this study were sourced from NASA Goddard Space Flight Center, specifically

from the Goddard Earth Sciences Data and Information Services Center (GES DISC), through the OMI/Aura Nitrogen Dioxide Total and Tropospheric Column 1-orbit L2 Swath 13x24 km V003 (OMNO2) (Krotkov et al., 2019). The OMI $NO_2$ algorithm retrieves estimated columns (total, tropospheric, and stratospheric) of nitrogen dioxide from OMI Level-1B calibrated radiance and irradiance data. The current version, v4.0, improves on the retrievals in prior versions in several significant ways. The OMNO2 algorithm aims to infer as much information as possible about atmospheric $NO_2$ from OMI measurements, with minimal dependence on model simulations.

The following filters of pixels are applied, following Dang et al. (2023): (1) nearly clear-sky scenes, with effective cloud fraction < 0.3; (2)surface reflectivity < 0.3; (3)solar zenith angles < $75°$; (4)viewing zenith angles < $65°$. In addition, we also ensure that the 'vcdQualityFlag' possesses an even integer value to align with recommended data quality standards. The air mass factor (AMF) converts the satellite-observed slant column density (SCD) into the vertical column density (VCD) using the $NO_2$ vertical profile ($n$) as follows:

$$VCD = \frac{SCD}{AMF(n)}. \tag{1}$$

AMF is mainly determined by atmospheric path geometry, $NO_2$ vertical profile, surface reflectance, and atmospheric radiative transfer properties. $NO_2$ exhibits optical thinness in the visible spectrum, facilitating the calculation of AMF (Lamsal et al., 2014). This calculation involves altitude-dependent scattering weights ($sw$) derived from a radiative transfer model and a priori profile shape of $NO_2$ as follows:

$$AMF = \frac{\sum_l sw \cdot x_a}{\sum_l x_a}, \tag{2}$$

where $x_a$ is the partial $NO_2$ column, $l$ denotes each layer, extending either from the ground to the tropopause or from the tropopause to the stratropopause. We updated the AMF of both tropopause and stratropopause separately using the $NO_2$ vertical profile simulated by GEOS-Chem in this study. The total column $NO_2$ concentration is calculated as the sum of the updated tropospheric vertical column density and stratospheric vertical column density. We regridded the total column amount of $NO_2$ to match the horizontal resolution of GEOS-Chem used in this study, which is 0.5 degrees latitude by 0.625 degrees longitude. Note that for comparison with OMI observations, we restrict our analysis to the time window between 13:00 and 14:00 local time, ensuring consistency with the OMI observation window.

## 2.1.3 VIIRS nighttime lights

Following the deployment of the most recent earth observation satellite series, the Joint Polar-orbiting Satellite System (JPSS), the inclusion of the Visible and Infrared Imaging Suite (VIIRS) Day Night Band on JPSS satellites has ushered in a remarkable advancement in low-light imaging capabilities (Elvidge et al., 2017), surpassing the capabilities of its predecessor, the Defense Meteorological Satellite Program (DMSP) Operational Linescan System (OLS) (Small et al., 2005). This study employed the V2.1 annual global VIIRS nighttime lights dataset for the year 2020 (Elvidge et al., 2021) to delineate urbanization patterns within the study area. The intensity of color corresponds to the level of urbanization, where brighter colors indicate higher urbanization levels. Building upon the findings of Shi et al. (2014), we adopted a threshold of 10 nW $cm^{-2}sr^{-1}$

for the urbanization which will be used as an input in LUBR observational operator as will be illustrated later. Accordingly, areas with annual nighttime light values exceeding 10 nW cm$^{-2}$sr$^{-1}$ were designated as urban regions.

## 2.2 GEOS-Chem Model

The chemical transport model employed in this study is GEOS-Chem, specifically version 13.4.0, available on the Zenodo (The International GEOS-Chem User Community, 2022). The model was driven by assimilated meteorological data from the NASA Global Modeling and Assimilation Office's Modern-Era Retrospective analysis for Research and Applications Version 2 (MERRA-2) as detailed in (Gelaro et al., 2017). It has a fully coupled aerosol–ozone–NOx–hydrocarbon chemistry representation (Park et al., 2004). We took the global simulation with a spatial resolution of 2° latitude by 2.5° longitude as the boundary conditions. The region of interest, constituting the nested modeling domain (0–55° N, 70–140° E), was characterized by a refined horizontal resolution of 0.5° latitude by 0.625° longitude, accompanied by 47 vertical layers. It is worth noting that the choice of this resolution is a common practice when using the GEOS-Chem classic version, striking a balance between computational complexity and computing power. In addition, it is also the finest resolution that remains computationally affordable when a substantial ensemble of models is required for data assimilation. The anthropogenic emissions over China are from the Multi-resolution Emission Inventory for China (Li et al., 2017). For anthropogenic emissions outside of China, we utilized data from the Community Emissions Data System (CEDS) inventory as detailed in (Hoesly et al., 2018). This inventory predominantly comprises aerosols, aerosol precursors, and reactive compounds. GEOS-Chem also integrates additional NOx emissions from diverse origins, encompassing soil and fertilizer use (Hudman et al., 2012), lightning (Murray et al., 2012), and shipping (Holmes et al., 2014). A preliminary 1-year spin-up simulation was conducted before the main simulation.

## 2.3 The dynamic urban/rural factor

To reveal the pronounced heterogeneity in the distribution of atmospheric pollutant concentrations within a grid, hourly ground-level NO$_2$ and PM$_{2.5}$ measurements obtained from China MEP were averaged by month to reveal discrepancies between urban and rural sites. Beyond the nationwide contrasts, we also examine variations within China's two most urbanized regions, namely the NCP and YRD. In Fig. 2, panels (a) and (b) depict the monthly distribution of ground-level NO$_2$ and PM$_{2.5}$ concentrations in urban and rural regions. The disparities in NO$_2$ and PM$_{2.5}$ levels between urban and rural areas within the NCP and YRD regions are narrower than the national scale. This observation aligns with the notion that urbanization contributes to a reduction in urban-rural disparities. The disparity between urban and rural NO$_2$ levels is notably greater than that observed for PM$_{2.5}$, a trend in agreement with the brief atmospheric lifespan of NO$_2$ and the long atmospheric residence time of PM$_{2.5}$.

The seasonality of Urban/Rural factors for NO$_2$ is also explored in Supplement Figure S3. It reveals that the Urban/Rural factor tends to be larger in spring and summer compared to autumn and winter, which contradicts the expected NO$_2$ lifetime. However, the difference is not significant and varies with changes in the research area. This could be attributed to the combined effects of factors like meteorological conditions, regional hotspots, human activities, biological sources, and topography. In addition, soil NO$_x$ emissions during summer can have a significant impact, particularly as they are a primary source for rural

areas (Lu et al., 2021). However, it is challenging to provide concrete evidence based on the available data because we cannot distinguish the sources of $NO_x$. Therefore, further refinement of the research area and consideration of multiple factors are necessary rather than concluding solely from ground observations.

It is important to note that the urban/rural factor must be dynamic, as it is determined not only by the level of urbanization but also by the level of pollution. Analysis of the three-year monthly dataset reveals robust linear correlations between urban and rural $NO_2$ as well as $PM_{2.5}$ concentrations across all scales, as depicted in Fig. 2. Consequently, we computed the dynamic urban/rural factors for $NO_2$ and $PM_{2.5}$ by dividing the monthly averaged urban concentrations by the monthly averaged rural concentrations. The national monthly factor exhibits a range of values from 1.4 to 1.8, with an average of 1.6. In the case of the NCP and YRD regions, their respective factors range from 1.2 to 1.7 and 1.0 to 1.4. The average values for NCP and YRD are 1.4 and 1.1, respectively.

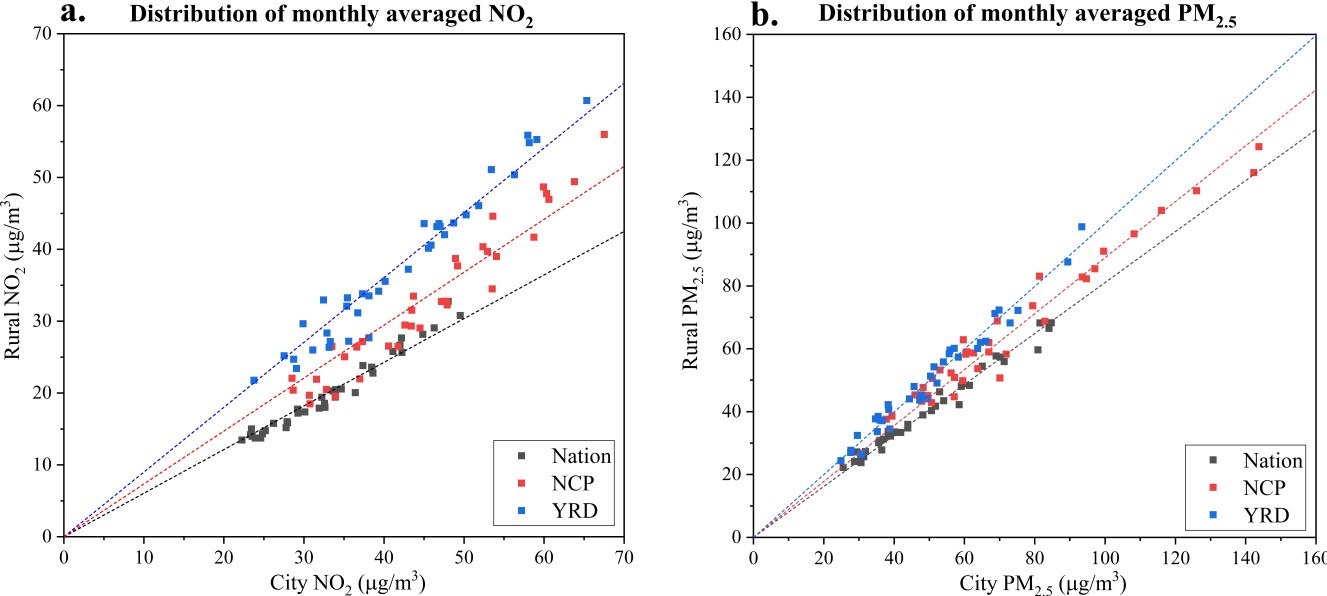

**Figure 2.** The distribution of monthly averaged ground observations between rural areas and urban areas. The national mean results and two clustered megacities - namely NCP and YRD - are shown in black, red, and blue rectangles, respectively. Panel a and Panel b present the results for $NO_2$ and $PM_{2.5}$, respectively.

## 2.4  The LUBR algorithm

The pseudocode outlining the LUBR algorithm is provided in Algorithm 1. The primary objective is to incorporate the urban and rural area proportions within each model grid, enhancing the representation of actual grid-level observations. Given the non-uniform distribution of monitoring stations, the VIIRS nighttime lights data boasts a fine resolution (Image Resolution: 15 arc seconds), enabling the differentiation between urban and rural regions. In this study, a threshold of 10 nW cm$^{-2}$sr$^{-1}$

is established for the VIIRS nighttime lights data to discriminate between urban and rural regions. Consequently, areas with values exceeding 10 nW cm$^{-2}$sr$^{-1}$ are classified as urban areas.

Each model grid cell, such as GEOS-Chem nested grids in this work, can be categorized into three possible types. The first pertains to grid cells exclusively encompassing urban sites, the second entails grid cells solely comprised of rural sites, and the third encompasses grid cells containing a combination of urban and rural sites. grid cells devoid of any sites fall beyond the scope of this study. Urban observation within a *U* and *Mix* type grid are computed either as the mean of urban sites or as the mean of rural sites multiplied by the urban/rural dynamic factor with a *R* grid. Similarly, rural observations from monitoring stations within each *R* and *Mix* grid are calculated either as the mean of rural sites or as the mean of urban sites divided by the urban/rural factor. Finally, the grid observations are calculated as the sum of urban observations multiplied by the proportion of urban area and rural observations multiplied by the proportion of rural area.

---

**Algorithm 1** The Land Use-Based Representation (LUBR) for gridded Observations

---

**Input:** Model grids $\{\text{grid}_i\}_{i=1}^I$, Observation data $\{\text{site}\}$, Annual VNL V2 data $\{\text{vnl}\}$, Urban/Rural factor $\{\text{factor}_n\}_{n=1}^N$

1: Initialize $I = \left(\frac{\text{lat\_max}-\text{lat\_min}}{0.5}+1\right) \times \left(\frac{\text{lon\_max}-\text{lon\_min}}{0.625}+1\right)$, threshold = 10, $n$ = month_begin (201501), $N$ = month_end (201712)

2: **for** $i = 1$ **to** $I$ **do**

3:     Find VNL data (vnl$_i$) from $\{\text{vnl}\}$ in grid$_i$

4:     Total area (TA$_i$) = COUNT(vnl$_i$)

5:     Urban area (UA$_i$) = COUNT(vnl$_i$ > threshold)

6:     Find observation data (site$_i$) from $\{\text{site}\}$ in grid$_i$

7:     **for** $n = 201501$ **to** $N$ **do**

8:         **if** COUNT(site$_i$) > 0 **then**

9:             **if** site$_i$ contains rural sites (sites$_R$) **then**

10:                 **if** site$_i$ contains urban sites (sites$_U$) **then**

11:                     Represented grid observation = $\text{MEAN}(\text{sites}_R) \times \frac{\text{TA}_i - \text{UA}_i}{\text{TA}_i} + \text{MEAN}(\text{sites}_U) \times \frac{\text{UA}_i}{\text{TA}_i}$

12:                 **else**

13:                     Represented grid observation = $\text{MEAN}(\text{sites}_R) \times \frac{\text{TA}_i - \text{UA}_i}{\text{TA}_i} + \text{MEAN}(\text{sites}_R) \times \text{factor}_n \times \frac{\text{UA}_i}{\text{TA}_i}$

14:                 **end if**

15:             **else if** site$_i$ contains urban sites (sites$_U$) **then**

16:                 Represented grid observation = $\frac{\text{MEAN}(\text{sites}_U)}{\text{factor}_n} \times \frac{\text{TA}_i - \text{UA}_i}{\text{TA}_i} + \text{MEAN}(\text{sites}_U) \times \frac{\text{UA}_i}{\text{TA}_i}$

17:             **end if**

18:         **else**

19:             No observations available, pass

20:         **end if**

21:     **end for**

22: **end for**

---

# 3 Result and discussion

Results and discussions in the following structure: We firstly performed the model validation in Section 3.1. Followed by the illustration of the discrepancy between observation and model in Section 3.2. Section 3.3 validates the accuracy of the LUBR operator in ground $NO_2$ observation model evaluation. Section 3.4 examines the benefit of using the LUBR operator.

## 3.1 Model validation

We validated the model by comparing daily simulations with ground observations collected from 2015 to 2017. The $R^2$ values for $NO_2$ and $PM_{2.5}$ were found to be 0.73 and 0.79, respectively. These results indicate that the model is capable of capturing the time variation in these pollutants to some extent. The NMB values for $NO_2$ and $PM_{2.5}$ were 57.68% and 20.4%, respectively, indicating that GEOS-Chem underestimates these pollutants compared to observations. Notably, the underestimation of $NO_2$ is more severe, with its NMB being more than twice that of $PM_{2.5}$.

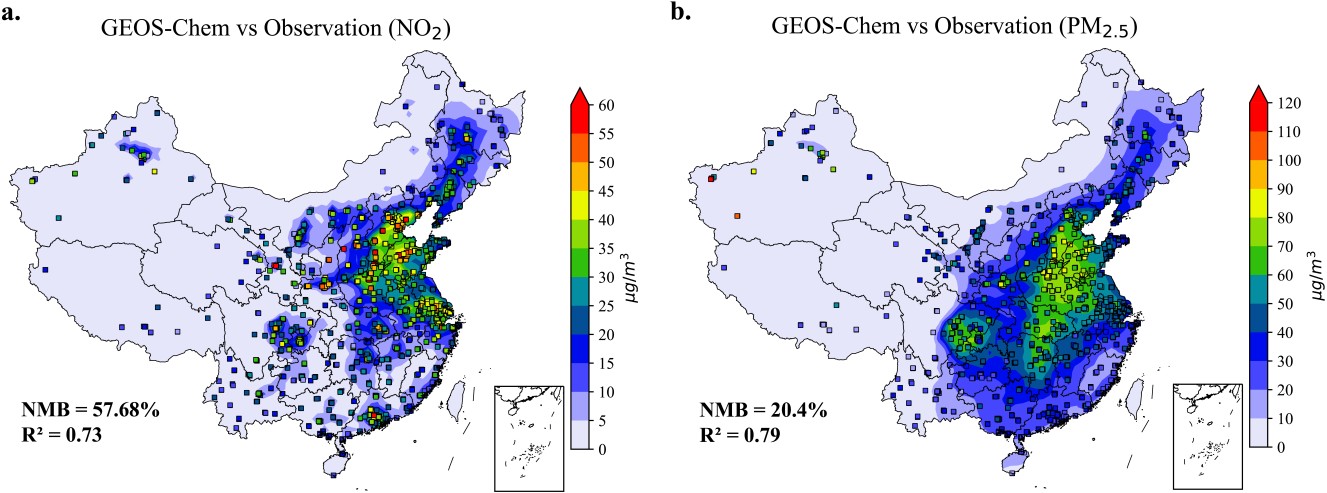

**Figure 3.** The model validation of GEOS-Chem for the simulation of ground $NO_2$ and $PM_{2.5}$. Panels a and b denote the three-year averaged ground $NO_2$ and $PM_{2.5}$ concentrations from GEOS-Chem simulation and ground observations, respectively. The NMB and $R^2$ for the $NO_2$ validation is 57.68% and 0.73. The NMB and $R^2$ for the $PM_{2.5}$ validation is 20.4% and 0.79.

## 3.2 The discrepancy between observation and model simulation

We averaged the model output between 13:00 and 14:00 local time for consistency with the timing of the Aura overpass for comparison with OMI observations. Fig. 4 shows inconsistent results when comparing the $NO_2$ simulation with ground-level $NO_2$ and OMI total column $NO_2$ measurements. In contrast to the irregular and sparse spatial distribution of ground observations, OMI observations offer high resolution and complete spatial coverage. Different from the ground-based stations that measure the pollutants in very surrounding areas, the OMI instrument quantified the mean status of the given pixel similarly

to the gridded numerical model simulation. Moving on to panels c and d, these show the spatial distribution of $NO_2$ column concentrations, averaged from 2015 to 2017, for the GEOS-Chem simulation and OMI observations, respectively. The black box corresponds to the NCP region, an area characterized by pronounced $NO_2$ pollution. For a clearer illustration of these disparities, panel g displays the scatter plot comparing the monthly $NO_2$ column concentrations from GEOS-Chem simulations

with the monthly OMI $NO_2$ observations. Panel h presents the same comparison focused on the NCP region. There is a slight underestimation by GEOS-Chem in terms of the total column $NO_2$ for the entire study area (panel g), with a negative normalized mean bias (NMB) of -23.53%, while a clear overestimation is observed in the NCP region (panel h), with a positive NMB value of 47.58%. The bias arises from uncertainties in both the retrieval algorithms of OMI products and the simulation of GEOS-Chem. For instance, Shah et al. (2020a) compared two OMI $NO_2$ retrievals, namely the European Quality Assurance for

Essential Climate Variables (QA4ECV) project's $NO_2$ ECV precursor product (Boersma et al., 2018) and the Peking University POMINO product version 2 (Lin et al., 2015), with GEOS-Chem. They found that GEOS-Chem overestimates OMI $NO_2$ when using the QA4ECV retrieval, while underestimating it when using POMINO. In addition, MEIC tends to overestimate NOx emissions in cities with lower industrial emission intensities or fewer industrial facilities (Wu et al., 2021), which may contribute to the overestimation of GEOS-Chem in these areas.

Panel a displays the GEOS-Chem ground-level $NO_2$ simulation, and panel b exhibits the corresponding observations from environmental monitoring stations. Similarly, panel e presents a scatter plot comparing monthly ground $NO_2$ concentrations between GEOS-Chem simulations and nationwide ground-level $NO_2$ observations. Panel f offers the same comparison, specifically focusing on the NCP region. In contrast, GEOS-Chem significantly underestimates $NO_2$ concentrations, evident in both the nationwide assessment (panel e, with a negative NMB value of -44.61%) and within the NCP region (panel f, with a neg-

ative NMB value of -25.5%). Evaluation or assimilation with these observational sources would inevitably mislead to higher $NO_2$ simulating levels. Similarly, Fig. 5 shows the results from the average of all hours from 2015 to 2017 rather than just 13:00-14:00, and the underestimation of GEOS-Chem exhibits a similar pattern. In the subsequent sections, we will concentrate on these monthly average ground observations for further discussion and evaluation of the LUBR algorithm. Notably, the ground observations in Fig. 4 and Fig. 5 used for comparison with the GEOS-Chem grid results are acquired by finding the

nearest observation point to each model grid cell, which is the most common method. We also conducted tests using the 'grid mean' method, but the results closely resembled those obtained with the 'nearest search' method.

The GEOS-Chem was validated to successfully reproduce the spatial distribution of the other pollutants like $PM_{2.5}$. Due to the inherently short lifetime of $NO_2$ results in the distribution of its concentrations within a GEOS-Chem grid exhibits pronounced heterogeneity, and hence the ground-based observation are not fairly comparable to the simulation via either the

30 'nearest search' or 'grid mean' operators as discussed in Section 2.3. Consequently, we suggest that a more effective approach is needed to accurately represent the true observations within each grid cell.

### 3.3 LUBR operator evaluation

In panels (f) and (h) of Fig. 4, inconsistencies between observations and GEOS-Chem simulations in the NCP are evident: GEOS-Chem underestimates ground-level $NO_2$ while overestimating $NO_2$ column concentrations. Although the bias between

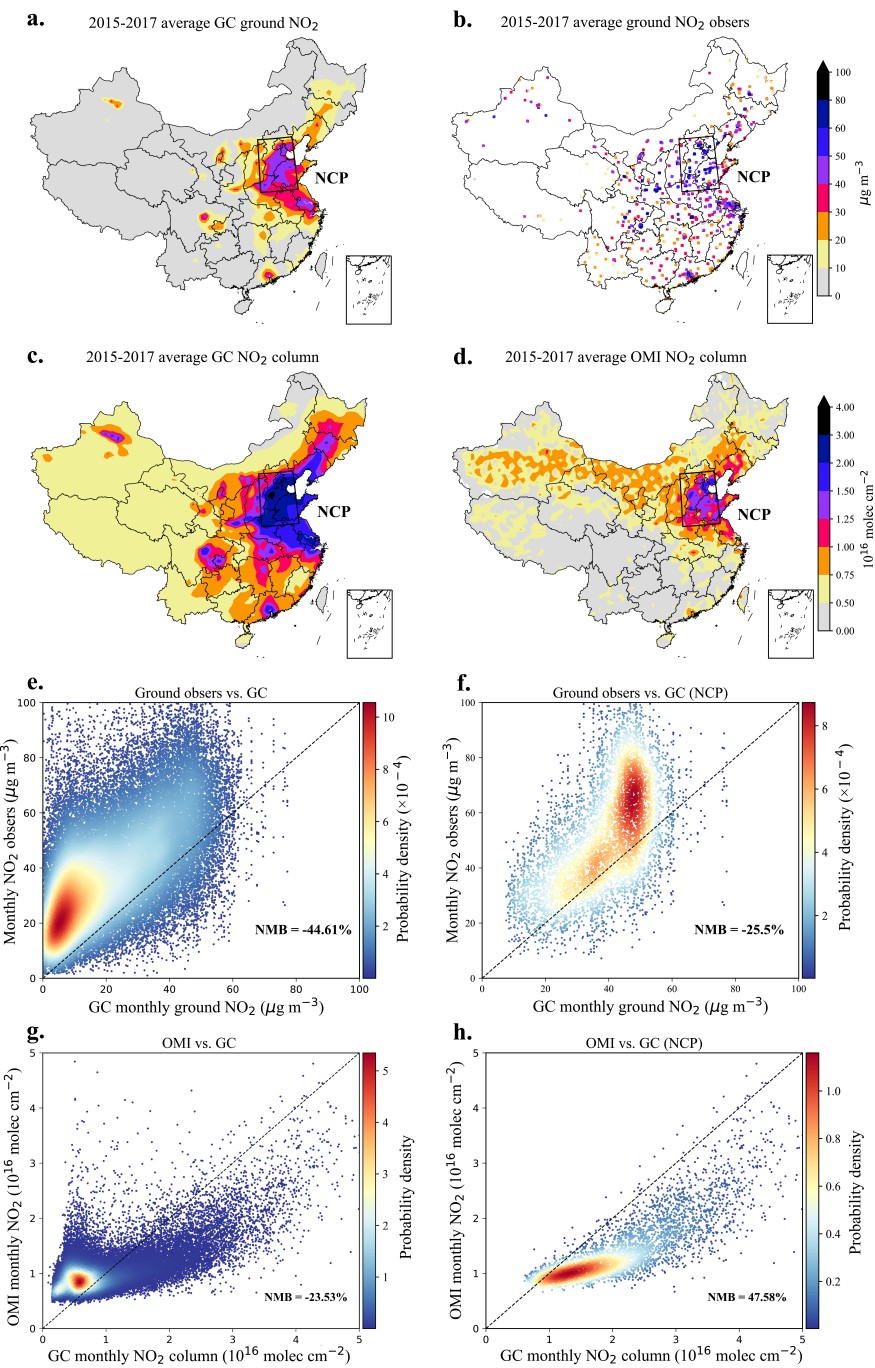

**Figure 4.** The inconsistency between the observations and GEOS-Chem simulations is evident. Panels a and b depict the spatial distribution of ground-level NO₂ from GEOS-Chem and monitoring sites, while panels c and d show the distribution of column-level NO₂ from GEOS-Chem and OMI. The NCP region, depicted by the black box, exhibits the most severe NO₂ pollution. The ground observations and model simulations represent the average conditions between 13:00 and 14:00 local time from 2015 to 2017. Panels e and g display scatter plots of the GEOS-Chem simulations and observations (monthly value), while panels f and h focus on the NCP region.

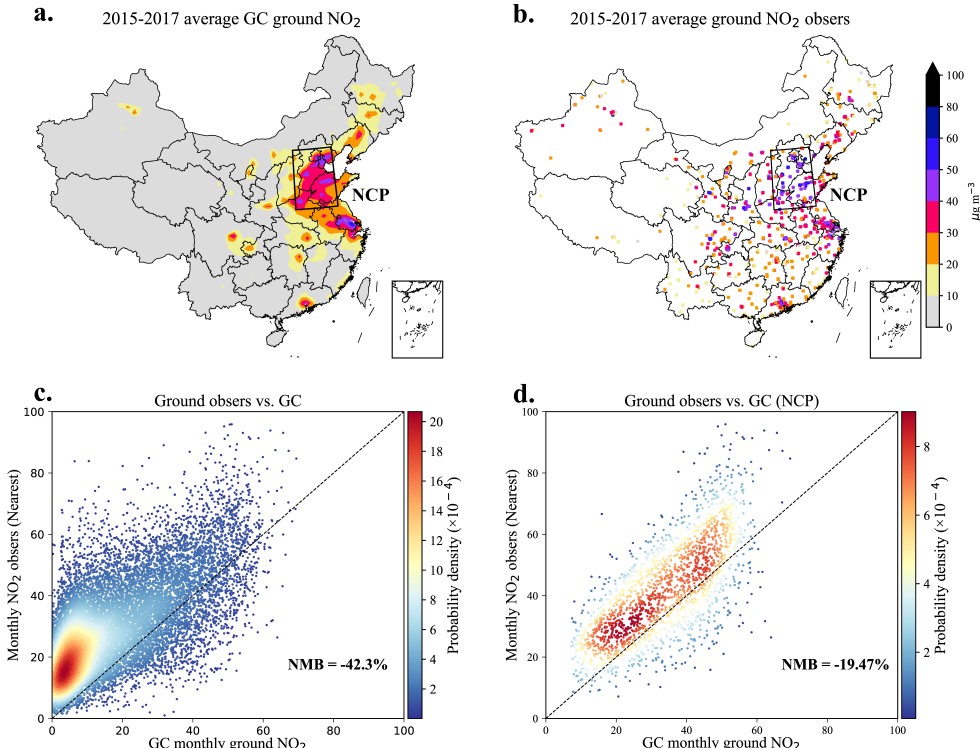

**Figure 5.** The spatial distribution and scatter plot of ground observations and GEOS-Chem simulations. Panels a and b depict the spatial distribution of ground-level $NO_2$ from GEOS-Chem and monitoring sites (average from 2015 to 2017). Panel e displays scatter plots of the GEOS-Chem simulations and ground observations (monthly value), while panel d focuses on the NCP region.

model and satellite observations may not align with the bias between model and ground-based observations, as satellites measure the column density of $NO_2$, which captures information not only from the surface but also from the troposphere and stratosphere, it's worth noting that considering the model is the same and is popular and reliable, they should not diverge in opposite directions. The spatial disparity between model simulations and ground observations can indeed result in poor

5    representation of grid cell observations, which is certainly one of the reasons for the differences. Therefore, our work primarily focuses on correcting the representativeness of ground observations and ensuring that the true correction direction closely aligns with the comparison results between model and satellite observations. Following the implementation of the LUBR observational operator, we present the corresponding scatter plots of monthly ground-level $NO_2$ concentrations from GEOS-Chem and observation using LUBR in Fig. 6. With the LUBR operator, the comparison against all ground stations now shows

10   our simulation did not overestimate ground-level $NO_2$ concentrations that much. The negative bias is remarkably reduced from -42.3% in Fig. 5(c) to -18.37% in Fig. 6(a). This result aligns more closely with the trend of comparing GEOS-Chem and OMI observations. Despite these improvements, most of the ground observations are located in urban areas sparsely, and cannot be directly compared to OMI observations, which provide comprehensive spatial coverage at the national scale. It is fairer to

compare the satellite-model evaluation against ground station-model calibration over the NCP region, where environmental monitoring stations are densely distributed (exceeding 215 sites). Here we observe a reversal of the results presented in panel f of Fig. 5 in panel d, where the NMB shifts from -19.47% to 6.58%. This change aligns the overall overestimation tendency of GEOS-Chem with the comparison of OMI (as shown in panel h of Fig. 4), where a positive NMB value is evident. The consistency of the OMI observations gives us the confidence to use valuable ground $NO_2$ observations in the model evaluation or assimilation with the LUBR operator.

## 3.4 Model evaluation

A comprehensive model evaluation is performed. Section 3.4.1 compares the gridded observations obtained from three different operators with GEOS-Chem simulations, focusing on spatially averaged results within the NCP and YRD regions. We also examine the annual ground-level $NO_2$ concentration patterns in the study area from 2015 to 2017 using three representation operators. This section also analyzes model under/overestimations in different regions after applying the LUBR method. Section 3.4.2 assesses the overall difference between the LUBR operator and other common methods using metrics such as normalized mean bias (NMB), root mean square error (RMSE), and mean absolute error (MAE). The formulas of these statistic matrics are given in Supplement Section 1.

### 3.4.1 Spatial and temporal result

To make the spatial comparison more reliable, we focus on two of the most developed megacities with dense environmental monitoring stations. Fig. 7 shows the distribution of spatially averaged outcomes of the grid observations using three operators with GEOS-Chem simulations in the NCP and YRD regions. In panel (a), the GEOS-Chem simulations persist in overestimating grid observations using both the 'grid mean' (aqua-green lower triangles) and 'nearest search' (blue triangles) opeartors in the NCP. And there are no significant differences between using the 'grid mean' and 'nearest search' operators. Conversely, in the same panel, GEOS-Chem simulations generally underestimate grid observations using the LUBR method (red dots), which is now consistent with the underestimation indicated by the OMI satellite measurements in Fig. 4, panel (h). Similar results are also evident in the YRD (panel b). This underscores the crucial importance of taking into account the representativeness of $NO_2$ observations.

In contrast to $NO_2$, the spatially averaged $PM_{2.5}$ grid observations obtained using the LUBR operator do not exhibit significant differences when compared to those obtained using the 'grid mean' and 'nearest search' operators in both the NCP (panel c) and YRD (panel d). This suggests that $PM_{2.5}$ does not exhibit a notable distinction between urban and rural areas, likely due to its long atmospheric lifetime, allowing for relatively uniform mixing in both urban and rural regions. Hence, distinguishing between urban and rural areas is less critical when representing $PM_{2.5}$ observations for grid resolutions similar to the one used in this study ($0.5° \times 0.625°$).

Fig. 8 shows the annual ground $NO_2$ concentration patterns in the study area from 2015 to 2017 using three different representation operators. The ground $NO_2$ levels from GEOS-Chem simulations (filled contours) generally capture the pollution pattern in the study area, characterized by high concentrations in the eastern region and low concentrations of pollutants

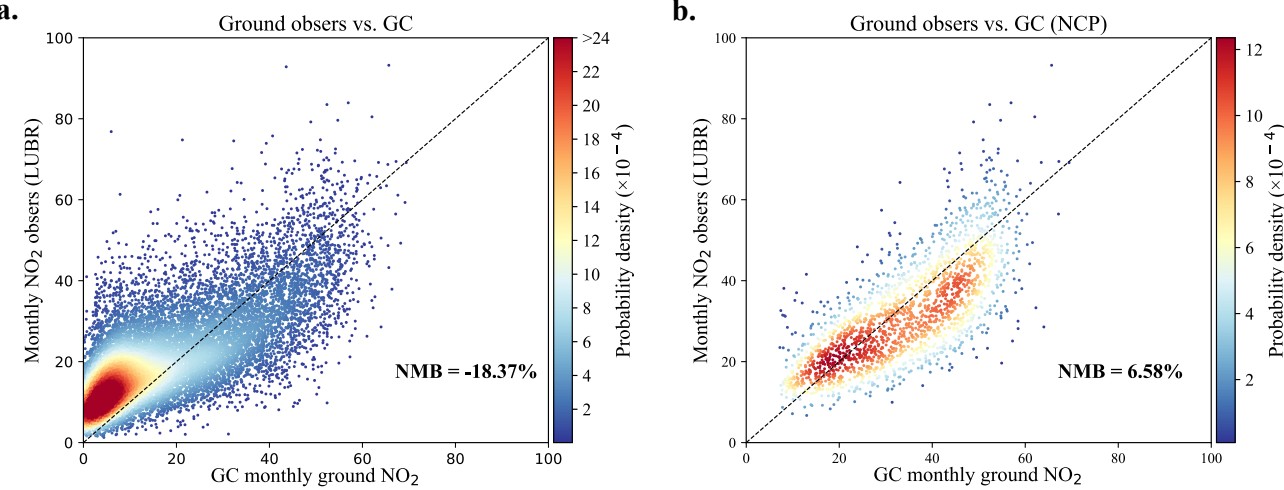

**Figure 6.** The scatter plot of ground-level $NO_2$ concentrations from GEOS-Chem and observed $NO_2$ concentrations using LUBR, based on monthly data spanning from 2015 to 2017. Panels a and b correspond to the results for the entire nation and the NCP region, respectively.

in the western areas. However, the comparisons against observations (colored squares) using 'grid mean' (panels b, e, h) and 'nearest search' (panels c, f, i) methods, show that GEOS-Chem simulations underestimate ground $NO_2$ concentrations in economically developed and severely polluted regions such as NCP and YRD, while overestimating ground $NO_2$ concentrations in less polluted regions. After achieving a more accurate representation of grid observations by incorporating information on urban-rural differences using the LUBR operator (panels a, d, g), the extent of underestimation by GEOS-Chem simulations in economically developed regions and overestimation in less polluted regions is mitigated.

For $PM_{2.5}$, as depicted in Supplement Figure S1, high $PM_{2.5}$ pollution levels from GEOS-Chem simulations are observed in eastern China and the Sichuan Basin (SCB; 28.5–31.5° N, 103.5–107° E). Despite the pronounced overestimation of $PM_{2.5}$ levels in the SCB region, in line with previous findings (Li et al., 2016; Fang et al., 2023), GEOS-Chem generally exhibits good agreement with actual $PM_{2.5}$ concentrations in the atmosphere. No substantial difference in the annual evaluation of GEOS-Chem is observed after applying the LUBR operator compared to the 'grid mean' and 'nearest search' operators. This is consistent with the previous spatial averaged results as the $PM_{2.5}$ does not exhibit significant urban/rural distinctions. Specific differences between using different operators in terms of statistical metrics will be presented later.

### 3.4.2 The statistical evaluation

As mentioned previously, our LUBR algorithm is applicable to calculate the mean status of atmospheric pollutants over three types of grids: $U$ containing only urban sites, $R$ with only rural sites, and *Mix* with both urban and rural sites. We will now discuss the distinctions observed within these three grid types on a national scale. Fig. 9 shows the statistical results of RMSE and MAE for the grid observation and GEOS-Chem simulations. The colors ice blue, rosy red, and cyan represent the

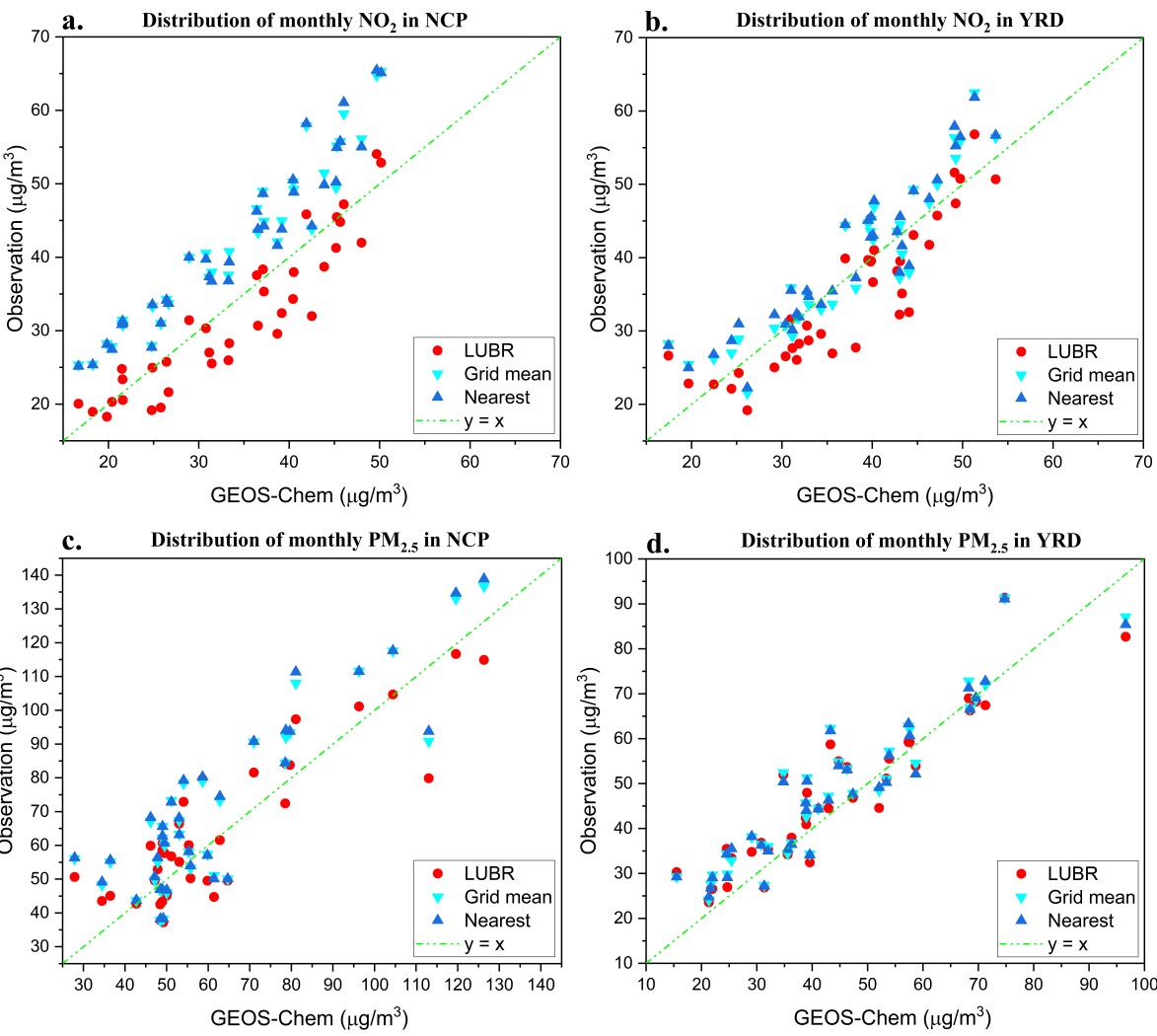

**Figure 7.** The distribution of spatially averaged results between ground observations and GEOS-Chem simulations. The results of LUBR, grid mean, and nearest search observational operators are represented by red dots, aqua-green lower triangles, and blue triangles, respectively. Panel a and b present the $NO_2$ results, while Panel c and d present the $PM_{2.5}$ results.

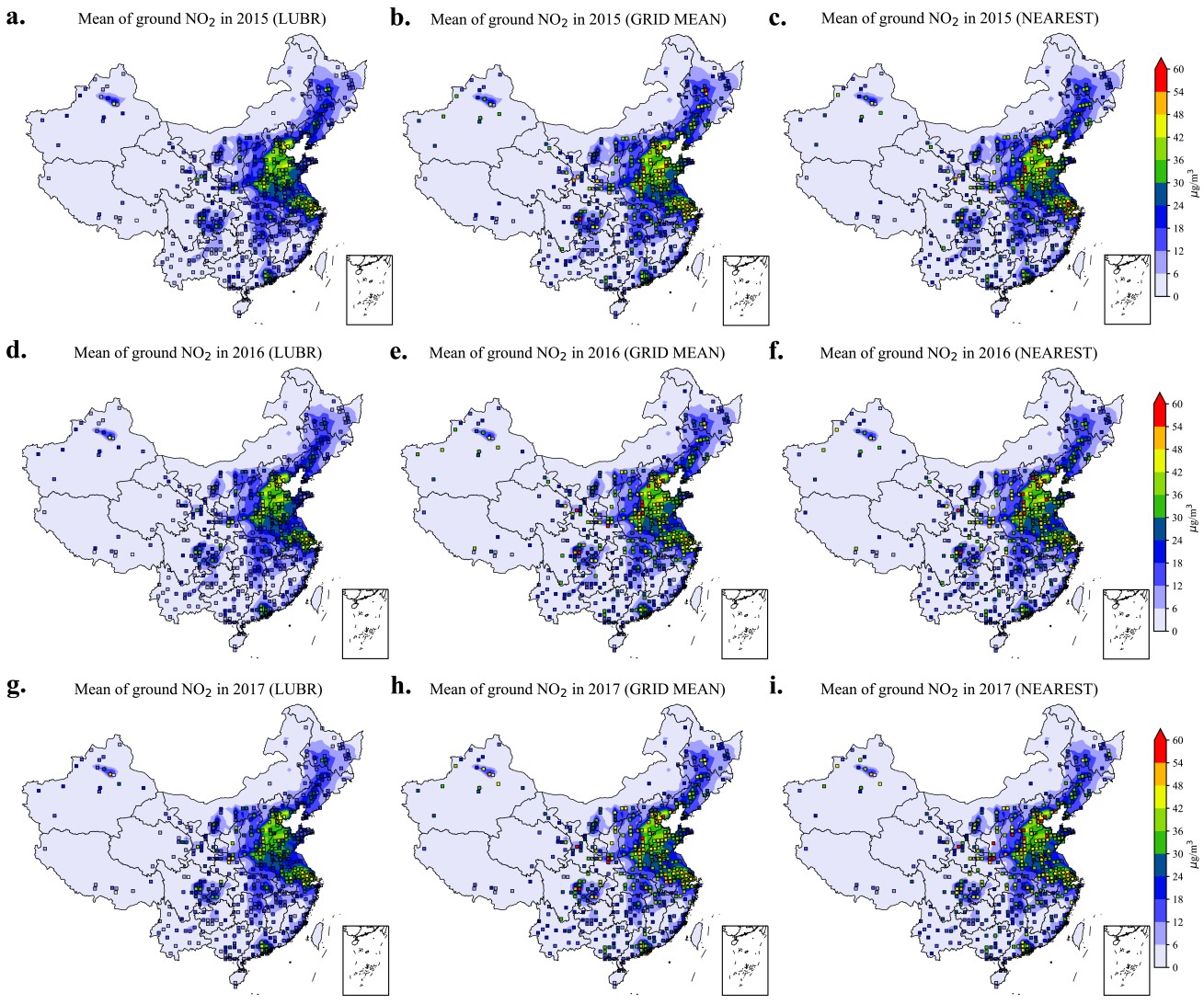

**Figure 8.** The annual averaged ground NO$_2$ from GEOS-Chem simulations (filled contours) and the represented observations of simulation grids (colored squares) from three operators. Panels a, d, and g present results using the LUBR operator to represent grid NO$_2$ concentrations for 2015, 2016, and 2017, respectively. Panels b, e, and h present results using the grid mean method. Panels c, f, and i present results using the nearest search method.

LUBR, 'nearest search', and 'grid mean' operators, respectively. The sample amounts of these three types of grids are shown in Supplement Figure S5. The gridded observations of $NO_2$ obtained from the 'nearest search' and 'grid mean' operators for grid types of *U* and *Mix* typically have higher RMSE and MAE values than the LUBR operators, indicating an inadequate representation of grid observation in terms of model evaluation. Remarkably, the utilization of the 'grid mean' operator demonstrates

significantly lower RMSE and MAE values compared to the 'nearest search' operator when applied to the *Mix* grid type. This underscores the critical importance of considering urban-rural information within grids and the 'grid mean' operator is better than the 'nearest search' operator in the grid type of *Mix* for model evaluation. However, in grid types of *U* and *R*, the minimal difference between these two operators is evident and easily explained, as these grid types lack urban-rural information within a single grid. The different statistics of the 'grid mean' operator and the 'nearest search' operator indicate that sites

within a specific grid cell can exhibit varying observations, particularly in grid type of *Mix*. While the differences are less pronounced due to the relatively low spatial heterogeneity of $PM_{2.5}$, similar trends are also noticeable in $PM_{2.5}$, as illustrated in Supplement Figure S4. During evaluation with GEOS-Chem results, the LUBR operator exhibits substantially lower RMSE and MAE values in grid types of *U* and *Mix*, as evident in Fig. 9. The RMSE and MAE of grid type of *U* decreased from 17.2 $\mu$g/m$^3$ and 14.5 $\mu$g/m$^3$ (the second-lowest results obtained from the 'grid mean' operator) to 10.1 $\mu$g/m$^3$ and 8.1 $\mu$g/m$^3$ after

applying the LUBR method. Similarly, the RMSE and MAE of grid type of *U* decreased from 13.5 $\mu$g/m$^3$ and 11.6 $\mu$g/m$^3$ to 11.7 $\mu$g/m$^3$ and 9.5 $\mu$g/m$^3$. Notably, the model bias in GEOS-Chem simulations remains unchanged; what we achieve is a reduction in the bias of grid observations. This also reveals that GEOS-Chem performs much better in the $NO_2$ simulation over China than our experience using the 'nearest search' or 'grid mean' observational operator. The LUBR operator can also, to some extent, aid in the evaluation of model simulations and observations for $PM_{2.5}$, as demonstrated in Supplement Figure

S4.

The LUBR operator demonstrates its most significant benefits in both $NO_2$ and $PM_{2.5}$ when applied to the grid type of *U*. This phenomenon can be attributed to the fact that grids composed solely of urban sites typically yield a larger volume of site observation data, thereby enhancing the reliability of the data. In contrast, the grid type of *Mix* often includes only one rural site, which is frequently situated close to urban areas due to rapid urbanization in China. These factors can lead to an

25 overestimation of actual rural $NO_2$ and $PM_{2.5}$ concentrations. Furthermore, we find minimal alterations in the grid type of *R* following the implementation of the LUBR operator for both $NO_2$ and $PM_{2.5}$. This lack of change can be attributed to the inherent characteristics of these grids, as they are typically situated in remote, non-urban regions and consist of just a single site. Consequently, the 'grid mean' and 'nearest search' operators produce identical results for these grids. Our evaluation of urban areas using Nighttime Light data similarly indicated the absence of significant urban areas within these grids. Therefore,

the effectiveness of the LUBR operator may be diminished in such locations.

Overall, the LUBR operator leads to a substantial enhancement in $NO_2$ grid observation representation, decreasing RMSE and MAE values by 34.5% and 37.0% when compared to the 'grid mean' operator and by 37.1% and 39.0% when compared to the 'nearest search' operator. The substantial bias in the observational operator not only misled the model evaluation but caused assimilation divergence as illustrated in our recent aerosol optical depth assimilation study (Jin et al., 2023).

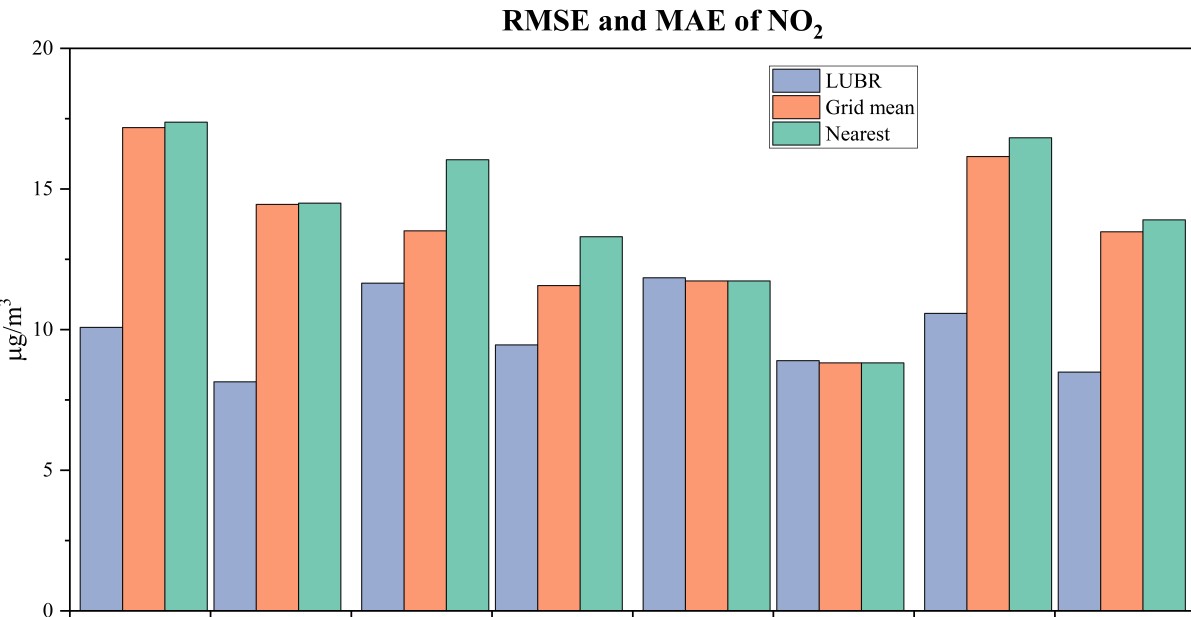

**Figure 9.** The comprehensive statistical results, including RMSE and MAE, demonstrate the distinctions of the gridded NO₂ observations compared to the GEOS-Chem simulations. The colors ice blue, rosy red, and cyan represent the LUBR, 'nearest search', and 'grid mean' operators, respectively. 'Urban,' 'Urban+Rural,' and 'Rural' categorize grids based on the presence of urban and rural sites. 'Urban' includes grids with exclusively urban sites, 'Urban+Rural' includes both urban and rural sites, and 'Rural' comprises grids with only rural sites. 'Total' aggregates results by calculating the average across all three categories.

## 4 Conclusion

The key finding of this work is the development of a new land-use-based observational operator (LUBR) that incorporates high-resolution urban-rural land-use data to improve the representativeness of ground monitoring observations when they are compared to air quality model simulations. This new operator is validated to provide a more accurate representation of grid-level observations from ground-level NO₂ measurements in the study area compared to traditional operators like 'nearest search' and 'grid mean'. It can lead to a change of up to 37% in RMSE and 39% in MAE in the context of model evaluation. The results highlight the importance of considering fine-scale intra-grid variability, especially for short-lived pollutants like NO₂ with large urban-rural gradients. This study provides an effective solution to address the spatial scale mismatch that has hindered robust model evaluation against ground-based monitoring data. The LUBR operator enables more accurate model evaluation and observational bias correction, which will benefit air quality modeling and predicting capabilities. The proposed operator is broadly applicable for model-observation evaluations of other atmospheric species with significant spatial heterogeneity within model grid cells. The LUBR algorithm, though effective, doesn't fully correct the representation error

as urban/rural sites cannot fully represent the average conditions of the entire urban/rural areas within this grid cell. Future endeavors could explore employing deep learning models to reveal the intricate relationship between the average conditions of grid cells and various factors beyond urban/rural sites, such as meteorology, climate, and land cover.

**Code and data availability**

The ground-based air quality monitoring observations are from the network established by the China Ministry of Environmental Protection and accessible via http://www.cnemc.cn/en/, the $NO_2$ data used in this paper is also archived on Zenodo (Fang, 2023b). The Land use information is also archived on Zenodo (Fang, 2023b). The Python source code of the LUBR observational operator is archived on Zenodo (Fang, 2023a). The simulation results of $NO_2$ from GEOS-Chem are archived on Zenodo (Fang, 2024).

**Financial support**

This work is supported by the National Natural Science Foundation of China [grant 42105109] and the Natural Science Foundation of Jiangsu Province (NO.BK20210664).

**Author contribution**

JJ conceived the study and designed the LUBR observational operator. LF wrote the code and carried out the evaluation. AS, KL, JX, WH, BL, HXL, LZ, SL, and HL provided useful comments on the paper. LF prepared the manuscript with contributions from JJ and all other co-authors.

**Competing interests**

The authors declare that they have no conflict of interest.

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
