# Peer review of "Observational operator for fair model evaluation with ground NO2 measurements"

_Geoscientific Model Development, 2023_

## Author Comment (AC1)

**Response to Referee #1:** We would like to thank the referee for the careful review throughout the paper that help to improve our paper.

Our Reply follows (*the referee's comments are in italics and blue*)

*General Comments*

*Representation error has posed a challenge in achieving consistent comparisons between models and ground-based observations. This issue arises because model grids are relatively coarse, whereas site-specific observations are locally representative, especially in heterogeneous environments targeting short-lived pollutants. This manuscript addresses this problem for NO2 by introducing a land-use-based representative (LUBR) observational operator, enabling the processed NO2 observations to better represent the means of 0.5x0.625 grid cells. This algorithm is proved effective for short-lived NO2 and is well evaluated in the paper. This method is helpful for accurately interpreting the bias between models and ground-based observations and is applicable to data assimilation research. I recommend this manuscript for publication once the issues outlined below are addressed.*

*Major Comments*

*An assumption underlying this LUBR algorithm is that observations from urban/rural sites can represent the average conditions of the entire urban/rural areas within this grid cell, which is not necessarily accurate. In other words, this algorithm only partially corrects the representation error, a point that needs clarifications.*

**Reply:** Thanks for the comments and this was not clearly explained in our previous version. It is indeed true that various factors, including meteorology, climate, and land cover, can influence representation errors. We have now added remarks to illustrate that there are also weaknesses of the LUBR algorithm and discussed prospects for future research in the ***Conclusion*** section. To clarify this, remarks are now added in page 19, line 8-12 by saying: "***The LUBR algorithm, though effective, doesn't fully correct the representation error as urban/rural sites cannot fully represent the average conditions of the entire urban/rural areas within this grid cell. Future endeavors could explore employing deep learning models to reveal the intricate relationship between the average conditions of grid cells and various factors beyond urban/rural sites, such as meteorology, climate, and land cover.***"

*In section 2.3, the authors compare modeled surface NO2 with ground-based observations, and modeled NO2 column with OMI observations. They note an inconsistent performance of model in simulating surface and column NO2. In their interpretation throughout the paper, satellite*

*observations are considered more representative, and the model-to-satellite bias is treated as the true bias for simulating NO2. However, this assumption may not be accurate, for reasons that are listed below. It is important to address these issues throughout the paper, although they do not compromise the paper's overall conclusion. (1) Satellite observations have their own representative issues and should be treated carefully. OMI provides observations only for the 1-2pm overpassing window and are most reliable under clear-sky conditions, when chemistry/meteorology might differ from monthly means. OMI retrievals require a prior NO2 profile shape, which can be a major source of retrieval error. A consistent comparison between OMI and GEOS-Chem requires the same sampling process for modeled NO2, and replacing the a prior NO2 profile shape in OMI retrieval with one simulated by GEOS-Chem. Only after these processes can the bias between the resampled model and reprocessed retrieval be considered the actual bias between the model and satellite observations. It appears that in this paper, the authors lack this preprocessing before determining the model-retrieval bias. This should be corrected. (2) Even with a correctly determined bias between model and satellite observations, it does not imply that this bias will align with the bias between model and ground-based observations. This is because satellites measure column density of NO2, capturing information not just from the surface but also from the troposphere and stratosphere (I assume they use total column density which includes stratospheric contribution – this needs to be clarified in the paper). Thus, it is entirely reasonable for column bias to differ from the surface bias. The authors should not regard column bias as the true bias for ground-level comparisons.*

**Reply:** Thanks for the in-depth comments, and we acknowledge that the model-to-satellite bias should not be considered as the true bias. Regarding your first point, we now recognize the significance of ensuring the same sampling time for modeled $NO_2$ and clear-sky conditions for OMI retrievals. Additionally, we now understand the importance of Aerosol Air Mass Factors in OMI retrievals and the necessity of substituting the priori $NO_2$ profile shape in OMI retrieval with one simulated by GEOS-Chem. It's also common in previous works to compare satellite data products with CTM directly. We would like to list some of these works directly using OMI standard products to compare with GEOS-Chem column concentrations below:

Wang, Yi, Jun Wang, Xiaoguang Xu, Daven K. Henze, Zhen Qu, and Kai Yang. "Inverse modeling of SO 2 and NO x emissions over China using multisensor satellite data–Part 1: Formulation and sensitivity analysis." Atmospheric chemistry and physics 20, no. 11 (2020): 6631-6650. https://acp.copernicus.org/articles/20/6631/2020/

Chen, Youfan, Lin Zhang, Daven K. Henze, Yuanhong Zhao, Xiao Lu, Wilfried Winiwarter, Yixin Guo et al. "Interannual variation of reactive nitrogen emissions and their impacts on PM2. 5 air pollution in China during 2005–2015." Environmental Research Letters 16, no. 12 (2021): 125004. https://iopscience.iop.org/article/10.1088/1748-9326/ac3695/meta

Wang, Zhe, Itsushi Uno, Keiya Yumimoto, Syuichi Itahashi, Xueshun Chen, Wenyi Yang, and Zifa Wang. "Impacts of COVID-19 lockdown, Spring Festival and meteorology on the NO2 variations in early 2020 over

China based on in-situ observations, satellite retrievals and model simulations." Atmospheric environment 244 (2021): 117972.  https://www.sciencedirect.com/science/article/pii/S1352231020307068

Considering the importance of updating NO2 profiles, we utilize the OMI L2 product instead of L3 to realize them. Besides, we resampled the GEOS-Chem modeled $NO_2$ to maintain consistency with the OMI local overpassing window (13:00-14:00 pm). Further details regarding the OMI product will be provided in your next major comment. To clarify this, comments remarking this are adding in page 6, line 7-24 by saying: "*The following filters of pixels are applied, following Dang et al. (2023): (1) nearly clear-sky scenes, with effective cloud5 fraction < 0.3; (2)surface reflectivity < 0.3; (3)solar zenith angles < 75°; (4)viewing zenith angles < 65°. In addition, we also ensure that the 'vcdQualityFlag' possesses an even integer value to align with recommended data quality standards. The air mass factor (AMF) converts the satellite-observed slant column density (SCD) into the vertical column density (VCD) using the $NO_2$ vertical profile (n) as follows:*"

$$VCD = \frac{SCD}{AMF(n)} \tag{1}$$

*AMF is mainly determined by atmospheric path geometry, $NO_2$ vertical profile, surface reflectance, and atmospheric radiative transfer properties. $NO_2$ exhibits optical thinness in the visible spectrum, facilitating the calculation of AMF (Lamsal et al., 2014). This calculation involves altitude-dependent scattering weights (sw) derived from a radiative transfer model and a priori profile shape of $NO_2$ as follows:*

$$AMF = \frac{\sum_l sw \cdot x_a}{\sum_l x_a} \tag{2}$$

*where xa is the partial $NO_2$ column, l denotes each layer, extending either from the ground to the tropopause or from the tropopause to the stratropopause. We updated the AMF of both tropopause and stratropopause separately using the $NO_2$ vertical profile simulated by GEOS-Chem in this study. The total column $NO_2$ concentration is calculated as the sum of the updated tropospheric vertical column density and stratospheric vertical column density. We regridded the total column amount of $NO_2$ to match the horizontal resolution of GEOS-Chem used in this study, which is 0.5 degrees latitude by 0.625 degrees longitude. Note that for comparison with OMI observations, we restrict our analysis to the time window between 13:00 and 14:00 local time, ensuring consistency with the OMI observation window.*" And in page 10, line 8-10 by saying: "*We averaged the model output between 13:00 and 14:00 local time for consistency with the timing of the Aura overpass for comparison with OMI observations.*" We also re-plotted the comparison figure to ensure that the model output is consistent with the sampling time of OMI, as shown in *Fig. 4*.

[Figure]

**Figure 4. The inconsistency between the observations and GEOS-Chem simulations is evident. Panels a and b depict the spatial distribution of ground-level NO₂ from GEOS-Chem and monitoring sites, while panels c and d**

*show the distribution of column-level NO₂ from GEOS-Chem and OMI. The NCP region, depicted by the black box, exhibits the most severe NO₂ pollution. The ground observations and model simulations represent the average conditions between 13:00 and 14:00 local time from 2015 to 2017. Panels e and g display scatter plots of the GEOS-Chem simulations and observations (monthly value), while panels f and h focus on the NCP region.*

Regarding your second point, we will not consider column bias as the true bias for ground-level comparisons but rather treat it as a point of comparison. Despite utilizing total column density of NO₂ data, which encompasses bias in both the troposphere and stratosphere, given the robust ability of NO₂ simulations in GEOS-Chem, we anticipate the overall tendency be similar. Remarks concerning this point are now added in page 11, line 3-9 by saying: "*The bias arises from uncertainties in both the retrieval algorithms of OMI products and the simulation of GEOS-Chem. For instance, Shah et al. (2020a) compared two OMI NO₂ retrievals, namely the European Quality Assurance for Essential Climate Variables (QA4ECV) project's NO₂ ECV precursor product (Boersma et al., 2018) and the Peking University POMINO product version 2 (Lin et al., 2015), with GEOS-Chem. They found that GEOS-Chem overestimates OMI NO₂ when using the QA4ECV retrieval, while underestimating it when using POMINO. In addition, MEIC tends to overestimate NOx emissions in cities with lower industrial emission intensities or fewer industrial facilities (Wu et al., 2021), which may contribute to the overestimation of GEOS-Chem in these areas.*" And in page 11, line 28-34 and page 13, line1-2 by saying: "*In panels (f) and (h) of Fig. 3, inconsistencies between observations and GEOS-Chem simulations in the NCP are evident: GEOS-Chem underestimates ground-level NO₂ while overestimating NO₂ column concentrations. Although the bias between model and satellite observations may not align with the bias between model and ground-based observations, as satellites measure the column density of NO₂, which captures information not only from the surface but also from the troposphere and stratosphere, it's worth noting that considering the model is the same and is popular and reliable, they should not diverge in opposite directions. The spatial disparity between model simulations and ground observations can indeed result in poor representation of grid cell observations, which is certainly one of the reasons for the differences. Therefore, our work primarily focuses on correcting the representativeness of ground observations and ensuring that the true correction direction closely aligns with the comparison results between model and satellite observations.*" And in page 13, line 6-7 by saying: "*This result aligns more closely with the trend of comparing GEOS-Chem and OMI observations.*"

*Additional information is needed regarding OMI product. Why was the OMAEROe product chosen? How does this product perform in comparison to ground based NO2 column observations and to other more popular OMI NO2 products, such as OMNO2 from NASA or the OMI product from KNMI?*

**Reply:** Thanks for point this, we apologize for misusing the OMI product, mistaking it for the one used in another study. In fact, the OMI product we used here is OMNO2 from NASA. Remarks concerning the detailed information are now added in supplement page 5, line 10-14 and page 6, line 1-6 by saying: "*Launched aboard the NASA EOS Aura satellite on July 15, 2004, OMI operates within a sun-synchronous ascending polar orbit. OMI conducts simultaneous measurements across a swath spanning 2600 km, partitioned into 60 Fields of View (FOVs). These FOVs range in dimension from approximately 13km x 24km near nadir to around 24km x 160km at the outermost FOVs. OMI provides observations only around 13:45(local time) overpassing window and is most reliable under clear-sky conditions. The NO2 total column concentrations utilized in this study were sourced from NASA Goddard Space Flight Center, specifically from the Goddard Earth Sciences Data and Information Services Center (GES DISC), through the OMI/Aura Nitrogen Dioxide Total and Tropospheric Column 1-orbit L2 Swath 13x24 km V003 (OMNO2) (Krotkov et al., 2019). The OMI NO$_2$ algorithm retrieves estimated columns (total, tropospheric, and stratospheric) of nitrogen dioxide from OMI Level-1B calibrated radiance and irradiance data. The current version, v4.0, improves on the retrievals in prior versions in several significant ways. The OMNO2 algorithm aims to infer as much information as possible about atmospheric NO$_2$ from OMI measurements, with minimal dependence on model simulations.*"

*I am curious if the urban/rural factor exhibits any seasonality, considering the longer lifetime of NO2 during winter compared to summer? Can soil NOx emissions during summer (dominant in rural areas?) influence the urban/rural factor? Please consider adding a discussion on this.*

**Reply:** Thanks for the suggestion. We consider it worthwhile to investigate the seasonality of urban/rural factors in future studies. Despite conducting searches for relevant papers, we could only find research focused on the city scale. Based on this study, we plotted the seasonal variation of NO$_2$ observations based on three years of ground observations from 2015 to 2017 in below *Fig. 0*. We observe that the urban/rural factor tends to be larger in spring and summer compared to autumn and winter, which is contrary to its expected lifetime. However, the difference is not significant and does not totally consistent when research area changed. We think it could be due to the combined effects of various factors such as meteorological conditions, regional hotpots, human activities, biological sources, and topography. Therefore, it may be necessary to refine the research area and consider multiple factors rather than conclude solely from the ground observations.

[Figure]

***Figure 0. The seasonal variation of the urban/rural factor of ground NO₂ concentrations. Each season is calculated from the average of 2015-2017 data, with blue, orange, and green representing the study areas of the Nation, NCP, and YRD respectively.***

For your second question, as reported by Lu et al., "The intensive nitrogen inputs to soil from fertilizer applications and nitrogen deposition lead to large soil NOx emissions via microbial processes, reaching 20% of the anthropogenic NOx emissions in summer over the NCP." These soil NOx emissions during summer can exert a significant influence, particularly as they constitute the main source for rural areas. However, it is challenging to provide concrete evidence based on the available data because we cannot distinguish the sources of NOx. Nevertheless, it does remind us of the importance of refining the Urban/Rural factor in the future.

Therefore, we would like to express our gratitude for the referee's insightful comments once again, as this direction appears promising. In the future, we plan to utilize more relevant data and employ advanced statistical models to conduct further in-depth research on the Urban/Rural factor.

*I don't see the point of figure 4, as it appears to convey ideas similar to those presented in figure 5 or 6. Please consider removing one of these figures to remain concise for evaluation section.*

**Reply:** Thanks for the comments. Figure 4 (now ***Fig. 6***) is intended to be directly compared with Figure 3(now ***Fig. 2***) as they convey a consistent message. On the other hand, Figure 5 and Figure 6 (now ***Fig. 7*** and ***Fig. 8***) present spatially averaged results, which may not contain as much information but provide a clearer view of the overall changes. Therefore, although they convey similar ideas, they serve different purposes, and we would like to retain them.

**Reference**

*Lu, X., Ye, X., Zhou, M. et al. The underappreciated role of agricultural soil nitrogen oxide emissions in ozone pollution regulation in North China. Nat Commun 12, 5021 (2021). https://doi.org/10.1038/s41467-021-25147-9*

---

## Author Comment (AC2)

**Response to Referee #2:** We would like to thank the referee for the careful review throughout the paper and the meaningful suggestion that helps to improve our paper.

Our Reply follows (*the referee's comments are in italics and blue*)

*General Comments*

*The authors develop an observational operator to improve the agreement between NO2 measurements from monitoring stations with low-resolution chemistry transport simulations. The operator uses VIIRS nighttime lights to estimate the urban and rural fraction in the grid cells and adjust the NO2 values based on fraction of urban/rural monitoring sites. The authors test the approach using GEOS-Chem simulations of China. They show that the operator reduces biases in grid cells compared to other operators (nearest search and grid means). The paper is within scope of GMD and provides some advances and tools that can be of interest to GMD readers. However, the paper is a few severe shortcomings that would need to be addressed.*

*The manuscript does not have a clear structure, which makes it difficult (and sometimes impossible) to follow the arguments of the authors. The "method" section includes many elements that would fit better to the "introduction" (motivation for the study) or the "result and discussion" section. For example, Section 2.3 is about the validation of GEOS-Chem with satellites and monitoring stations. Likewise, the result section includes element that would better fit into the method section. The manuscript is not very concise with many statements repeated at different places. The language is sometimes difficult to understand and inappropriate for a scientific paper (e.g.,: "achieving perfection", "intriguing phenomena", "truth revealed").*

**Reply:** We have reorganized the structure of this manuscript accordingly and polished the English languages thoroughly. The structure is revised as follows. ***Section 2.1*** and ***Section 2.2*** describe the study domain, observations, and model used. Details on the urban/rural factors and the LUBR algorithm are provided in ***Section 2.3*** and ***Section 2.4***. ***Section 3.1*** first provides the model validation, followed by the revelation of discrepancies between observations and model simulations in ***Section3.2***. The comprehensive evaluation of LUBR operator is then presented in ***Section 3.3***. The OMI data is discussed only in ***Section3.2*** and ***Section 3.3***. Note that we have distinguished the ground observations and ground simulations into two types. One type is averaged only from 13:00-14:00 local time, which is specifically for comparison with OMI data. The other type is averaged from all times to evaluate the LUBR algorithm. After evaluation of LUBR, ***Section 3.4*** discusses the spatial and temporal model evaluations of NO$_2$ and PM$_{2.5}$ pollutants either using LUBR or using the traditional grid mean/nearest search methods. Statistical

metrics that specifically focus on these observational operators, quantifying their performance, are also analyzed.

*The authors make heavy use of the term "model calibration". I have not seen the term anywhere before nor could I find a reference where the term is explained. The authors cite Zhu et al. (2021), who does not use the term, and Kalnay (2002), who writes that "...representativeness errors can be systematic or random. Systematic errors and biases should be determined by calibration or other means such as time averages." (p199). I interpret this as calibration of the observations and not the model. The suggested LUBR operator is also applied to adjust the observations and not the model. I think it would be a good idea to use a more common term.*

**Reply:** It is indeed that the LUBR operator is specifically designed for observations. With that, we can fairly compare the model simulations against the observations. We now use a more general term '***model evaluation***' instead of '***model calibration***' and the title is now changed to "***Observational operator for fair model evaluation with ground NO₂ measurements***".

*The validation of the GEOS-Chem with OMI NO2 in Section 2.3 is not reproducible from the provided information. The authors mention their previous study without adding a reference. OMI observations are mentioned for the first time in Section 2.3. It is not clear which OMI product (NASA, Dutch or a custom product) is actually used for validation. However, when comparing OMI NO2 columns data with model simulations it is necessary to update the air mass factors (AMFs) using the averaging kernels or scattering weights provided by the product. OMI NO2 can be biased for various reasons (NO2 profile shapes, surface reflectance and aerosols), which are not considered in the validation. Importantly, the OMI standard products tend to underestimate NO2 columns in China, which can explain the discrepancy (e.g., Lin et al. 2015, https://acp.copernicus.org/articles/15/11217/2015/).*

**Reply:** Thanks for the comments. The detailed information about OMI product is given in ***Section 2.1.2***. The NO₂ simulation from GEOS-Chem is uploaded in ***Zenodo*** to ensure reproducibility now. Comments remarking this are adding in page 5, line 10-14 and page 6, line 1-6 by saying: "***Launched aboard the NASA EOS Aura satellite on July 15, 2004, OMI operates within a sun-synchronous ascending polar orbit. OMI conducts simultaneous measurements across a swath spanning 2600 km, partitioned into 60 Fields of View (FOVs). These FOVs range in dimension from approximately 13km x 24km near nadir to around 24km x 160km at the outermost FOVs. OMI provides observations only around 13:45(local time) overpassing window and is most reliable under clear-sky conditions. The NO₂ total column concentrations utilized in this study were sourced from NASA Goddard Space Flight Center, specifically from the Goddard Earth Sciences Data and Information Services Center (GES DISC), through the OMI/Aura Nitrogen Dioxide Total and Tropospheric Column 1-orbit L2 Swath 13x24 km***

*V003 (OMNO2) (Krotkov et al., 2019). The OMI NO₂ algorithm retrieves estimated columns (total, tropospheric, and stratospheric) of nitrogen dioxide from OMI Level-1B calibrated radiance and irradiance data. The current version, v4.0, improves on the retrievals in prior versions in several significant ways. The OMNO2 algorithm aims to infer as much information as possible about atmospheric NO₂ from OMI measurements, with minimal dependence on model simulations.*" And in page 10, line 8-10 by saying: "*We averaged the model output between 13:00 and 14:00 local time for consistency with the timing of the Aura overpass for comparison with OMI observations.*" And in "*Code and data availability*" by saying: "*The simulation results of NO₂ from GEOS-Chem are archived on Zenodo (Fang, 2024).*"

Since the "previous work" is still incomplete at the moment, the sentence is poorly constructed, and thus we have removed it. We now recognize the significance of ensuring the same sampling time for modeled NO₂ and clear-sky conditions for OMI retrievals. Additionally, we now understand the importance of Aerosol Air Mass Factors in OMI retrievals and the necessity of substituting the priori NO₂ profile shape in OMI retrieval with one simulated by GEOS-Chem. Therefore, we utilize the OMI L2 product instead of L3 to realize them. Comments remarking this are adding in page 6, line 7-24 by saying: "*The following filters of pixels are applied, following Dang et al. (2023): (1) nearly clear-sky scenes, with effective cloud5 fraction < 0.3; (2)surface reflectivity < 0.3; (3)solar zenith angles < 75°; (4)viewing zenith angles < 65°. In addition, we also ensure that the 'vcdQualityFlag' possesses an even integer value to align with recommended data quality standards. The air mass factor (AMF) converts the satellite-observed slant column density (SCD) into the vertical column density (VCD) using the NO₂ vertical profile (n) as follows:*"

$$VCD = \frac{SCD}{AMF(n)} \tag{1}$$

*AMF is mainly determined by atmospheric path geometry, NO₂ vertical profile, surface reflectance, and atmospheric radiative transfer properties. NO₂ exhibits optical thinness in the visible spectrum, facilitating the calculation of AMF (Lamsal et al., 2014). This calculation involves altitude-dependent scattering weights (sw) derived from a radiative transfer model and a priori profile shape of NO₂ as follows:*

$$AMF = \frac{\sum_l sw \cdot x_a}{\sum_l x_a} \tag{2}$$

*where xa is the partial NO₂ column, l denotes each layer, extending either from the ground to the tropopause or from the tropopause to the stratropopause. We updated the AMF of both tropopause and stratropopause separately using the NO₂ vertical profile simulated by GEOS-Chem in this study. The total column NO₂ concentration is calculated as the sum of the updated tropospheric vertical column density and stratospheric vertical column density. We regridded the total column amount of NO₂ to match the horizontal resolution of GEOS-Chem used in this study, which is 0.5 degrees latitude*

*by 0.625 degrees longitude. Note that for comparison with OMI observations, we restrict our analysis to the time window between 13:00 and 14:00 local time, ensuring consistency with the OMI observation window.*"

We should not consider the OMI product as providing true column observations, as uncertainties from different retrieval algorithms can vary significantly. Regarding your last comment concerning "the OMI standard products tend to underestimate NO2 columns in China," we will address it in the following comment. After updating the air mass factors using the scattering weights provided by the product, we re-plotted the comparison figure as shown in *Fig. 4*.

[Figure]

**Figure 4. The inconsistency between the observations and GEOS-Chem simulations is evident. Panels a and b depict the spatial distribution of ground-level NO₂ from GEOS-Chem and monitoring sites, while panels c and d show the distribution of column-level NO₂ from GEOS-Chem and OMI. The NCP region, depicted by the black**

*box, exhibits the most severe NO₂ pollution. The ground observations and model simulations represent the average conditions between 13:00 and 14:00 local time from 2015 to 2017. Panels e and g display scatter plots of the GEOS-Chem simulations and observations (monthly value), while panels f and h focus on the NCP region.*

We also re-plotted ***Fig. 5***, which only contains ground observations averaged over the entire month rather than 13:00-14:00, to avoid direct comparison with OMI observations. ***Fig.6***, utilizing the LUBR algorithm, is solely used to compare with ***Fig. 5*** to maintain consistency.

[Figure]

*Figure 5. The spatial distribution and scatter plot of ground observations and GEOS-Chem simulations. Panels a and b depict the spatial distribution of ground-level NO₂ from GEOS-Chem and monitoring sites (average from 2015 to 2017). Panel e displays scatter plots of the GEOS-Chem simulations and ground observations (monthly value), while panel d focuses on the NCP region.*

*The authors conclude from their validation with OMI data that the GEOS-Chem simulations are overestimated, which, as written above, might be wrong. However, this is assumption is used to argue the improvements in the RMSE and MAE from the LUBR operator. Given this importance in the paper, the potential problem with the OMI data need to be addressed and the impact on the statistical evaluation (Section 3.2.2) reassessed.*

**Reply:** Followed you last comment, we believe the bias can be attributed to both OMI products especially the retrieval algorithms and the model simulations which have many unavoidable uncertainties. In this

work, the comparison result was only based this simulation results of GEOS-Chem and the specific OMI standard product we used. To clarify this, comments now are adding in page 11, line 3-9 by saying: "***The bias arises from uncertainties in both the retrieval algorithms of OMI products and the simulation of GEOS-Chem. For instance, Shah et al. (2020a) compared two OMI $NO_2$ retrievals, namely the European Quality Assurance for Essential Climate Variables (QA4ECV) project's $NO_2$ ECV precursor product (Boersma et al., 2018) and the Peking University POMINO product version 2 (Lin et al., 2015), with GEOS-Chem. They found that GEOS-Chem overestimates OMI $NO_2$ when using the QA4ECV retrieval, while underestimating it when using POMINO. In addition, MEIC tends to overestimate $NO_x$ emissions in cities with lower industrial emission intensities or fewer industrial facilities (Wu et al., 2021), which may contribute to the overestimation of GEOS-Chem in these areas.***"

For you second point, it's totally true this assumption should not be used to argue the improvements. In "***Section 3.4 Model evaluation***", the comparisons no longer involve $NO_2$ column concentrations but instead focus on the three ground operators for ground $NO_2$. I think you mean the "***Section 3.3 LUBR operator evaluation***" should not based on this assumption either. So we reassess the corresponding content in this Section in page 11, line 28-34 and page 13, line1-2 by saying: "***In panels (f) and (h) of Fig. 3, inconsistencies between observations and GEOS-Chem simulations in the NCP are evident: GEOS-Chem underestimates ground-level $NO_2$ while overestimating $NO_2$ column concentrations. Although the bias between model and satellite observations may not align with the bias between model and ground-based observations, as satellites measure the column density of $NO_2$, which captures information not only from the surface but also from the troposphere and stratosphere, it's worth noting that considering the model is the same and is popular and reliable, they should not diverge in opposite directions. The spatial disparity between model simulations and ground observations can indeed result in poor representation of grid cell observations, which is certainly one of the reasons for the differences. Therefore, our work primarily focuses on correcting the representativeness of ground observations and ensuring that the true correction direction closely aligns with the comparison results between model and satellite observations.***" And in page 13, line 6-8 by saying: "***This result aligns more closely with the trend of comparing GEOS-Chem and OMI observations. Despite these improvements, most of the ground observations are located in urban areas sparsely, and cannot be directly compared to OMI observations, which provide comprehensive spatial coverage at the national scale.***"

*The references often do not support the statement in the manuscript (see specific comments for examples). The authors should check there references especially when making general statements. Several links are not working.*

**Reply:** Thank you for reminding and we will recheck seriously.

*The authors provide Python code and NO2 measurements, but no GEOS-Chem fields. It is therefore not possible to test the new operator even on a small dataset. It would be great if at least a test dataset can be provided that demonstrates the application of the operator. In the current version, the datasets are very small and it would easier to add them as a supplement directly to the manuscript.*

**Reply:** Certainly, we are willing to provide the GEOS-Chem fields. Since the model outputs are quite large, we initially extract and calculate the ground $NO_2$ and column $NO_2$ results and upload them to Zenodo. Additionally, if someone wishes to conduct the simulation themselves, we are happy to provide the model setting file upon request. Comments remarking this now are adding in "***Code and data availability***" by saying: "***The simulation results of NO₂ from GEOS-Chem are archived on Zenodo (Fang, 2024).***"

*Specific comments*

*P2L23: Many measurement techniques (incl. remote sensing) are not direct measurements.*

**Reply:** Right, we will remove "***direct***".

*P2L29: The references do not support the statement that "satellite remote sensing [...] made it possible to observe near-surface air pollutant [...] from space". Zhang et al. 2020 describes the EMI NO2 retrieval algorithm and Jin et al. 2023a describes top-down NH3 emission estimates based on IASI. Possible references are Xu et al. 2019 (https://doi.org/10.1016/j.scitotenv.2018.11.125), Kim et al. 2021 (https://doi.org/10.1016/j.rse.2021.112573) and many other studies.*

**Reply:** We apologize for the misuse of citations, and we will incorporate your suggestions to replace them.

*P3L1ff: The rest of this section discusses monitoring stations, but what is the role of the satellite observations introduced in the previous paragraph?*

**Reply:** Thanks for the comment. We do miss to introduce the role of satellite observations here. They are characterized as full and fine spatial resolution compared with ground observations. Comments remarking this will be added in page 3, line 3-6 by saying: "***Observations from satellites typically have finer spatial resolution than model simulations, so the comparison between them is less affected by spatial scale disparity. Conversely, ground observations are sparse and uneven, making it more challenging to compare them with model gridded simulations.***"

*P3L4ff: A third approach is only using stations that spatially and temporally representative for the model grid.*

**Reply:** Thanks for the comment. The third approach sounds promising and can effectively represent the observations of model grid cells. The 'grid mean' and 'nearest search' methods are widely used because they are simple to implement in practice. However, there is no standard definition for determining the extent to which monitoring stations can represent model grids. Additionally, this method may result in unavoidable loss of valuable ground observations. Comments remarking this will be added in page 3, line 9-11 by saying: "***A third approach could be only using monitoring stations that are spatially and temporally representative for the model grid cells. However, there is no standard definition for determining the extent to which monitoring stations can represent model grids. Additionally, this method may result in unavoidable loss of valuable ground observations.***"

*P3L15: What does it mean when an approach is "unfair"?*

**Reply:** We also think "***unfair***" is not appropriate here, and the sentence now is revised as "***The aforementioned two commonly used methods for model evaluation can potentially cause large representative errors of observations.***" in page 3, line 20-21.

*P3L16f: A more common interpretation of grid cells is that they represent the mean state in a region. I do not think that Tessum et al. (2017) claim that a grid cell corresponds to distinct spatial location.*

**Reply:** We will take your advice and comments remarking this are added in page 3, line 21-22 by saying: "***The CTMs divide the atmosphere into a series of horizontal and vertical grid cells. Each grid cell represents the mean state in a specific region (Yan et al., 2016).***"

*P3L17f: A spatial resolution of 0.5° is not very high for regional chemistry simulations, which nowadays are often run at about 10 km simulations.*

**Reply:** We admit that if the model resolution is about 10 km or even small would greatly reduce the observational representative error. However, when considering relatively large regions or long-time simulations, or when using ensemble models for data assimilation, computational costs increase rapidly. Thus, we must strike a balance between computational complexity and computing power. We explained it in page 7, line 10-13 by saying: "***It is worth noting that the choice of this resolution is a common practice when using the GEOS-Chem classic version, striking a balance between computational complexity and computing power. In addition, it is also the finest resolution that remains computationally affordable when a substantial ensemble of models is required for data assimilation.***"

*P3L23ff: Wu et al. 2021b do not claim that anthropogenic NO2 emissions primarily occur in the troposphere. They actually write on the spatial variability NO2 emissions: "traffic-related pollutants in*

*urban environments can vary substantially within a few meters (Pattinson et al., 2014; Targino et al., 2016)."*

**Reply:** Thanks for the comment. The reference has now been changed to "***Meng Li, Huan Liu, Guannan Geng, Chaopeng Hong, Fei Liu, Yu Song, Dan Tong, Bo Zheng, Hongyang Cui, Hanyang Man, Qiang Zhang, Kebin He, Anthropogenic emission inventories in China: a review, National Science Review, Volume 4, Issue 6, November 2017, Pages 834–866, https://doi.org/10.1093/nsr/nwx150,***" as this paper provides a table to support the statement.

*P4L19f / Figure 1: The country borders in the figure do not follow the rules for GMD papers: "Please adhere to United Nations naming conventions for maps used in your manuscript. In order to depoliticize scientific articles, authors should avoid the drawing of borders or use of contested topographical names." (https://www.geoscientific-model-development.net/submission.html#mapsaerials)*

**Reply:** We will replace the country name with "***the study area***" to avoid any potential political issues.

*P5L1ff: The first sentences of the paragraph repeat partly the introduction.*

**Reply:** We will remove the first sentences to remain concise.

*P6L16f: The model validation with monitoring stations should not be in the supplement, but in the main part of the paper.*

**Reply:** Sure, we will move it to ***Section 3.1 Model validation***.

*P7L13f: The vertical profiles of NO2 and PM2.5 should be quite different, exactly because of their different lifetimes. Therefore, you cannot argue that incorrect vertical profiles cannot be the reason for the differences.*

**Reply:** Yes, the corresponding sentences are removed now.

*P7L23f: Do the stations are grouped by GEOS-Chem grid cells? Please also clarify if Figure 3 depicts GEOS-Chem or measurement stations.*

**Reply:** The urban/rural factor is solely based on ground measurements obtained from the China MEP. While refining the factors to each grid cell would be meaningful, many grid cells lack ground observations. Therefore, we derive the factor based on regions to obtain a relatively coarse estimate. We have clarified that ***Fig. 3*** (now ***Fig. 2***) depicts ground observations in the caption.

*P7L29ff: The motivation and meaning of the "dynamic urban/rural factor" needs to be explained in more details here.*

**Reply:** We will stress the importance of urban/rural factor in page 7, line 29-30 by saying: "***It is important to note that the urban/rural factor must be dynamic, as it is determined not only by the level of urbanization but also by the level of pollution.***"

*P10L5ff: Since it is unclear if OMI NO2 observations are not biased (see previous comment), I think the statement that the simulations overestimate atmospheric NO2 needs to be reassessed. The statement that the "truth revealed by the OMI comparison" is also very bold and should be rephrased.*

**Reply:** Yes, indeed. We have reassessed it in a different way and answered it in the major comments.

*P11L10f: The model calibration should be explained in the method section.*

**Reply:** We have taken more general term "***model evaluation***" to replace "***model calibration***".

*P15L15ff: Please provide an explanation why "grid mean" and "nearest search" have different statistics.*

**Reply:** We do forget to explain it more explicitly in previous version. We will explain it in page 18, line 5-6 by saying: "***The different statistics of the 'grid mean' operator and the 'nearest search' operator indicate that sites within a specific grid cell can exhibit varying observations, particularly in grid type of Mix.***"

*Technical correction (incomplete)*

*P3L27 ("grid pattern"): Do you mean grid cell?*

**Reply:** Yes, and replaced for more common language.

*P5L9 (also P9L8ff): grids -> grid cells*

**Reply:** Corrected.

*Figure 1: "lightning" -> "night lights" and "blue" -> "purple"*

**Reply:** Corrected.

*P6L9f: The statement repeats P3L19f*

**Reply:** The statements in '***P6L9ff***' are not appropriate in '***Materials and methods***' section and are removed.

*P7L22: "locales" -> "sites"*

**Reply:** Replaced.

*Figure 3a: The colors of the dashed black and blue lines are reversed.*

**Reply:** Corrected.

*References: Please check your references. Several of the links in the references are badly formatted or at not working.*

**Reply:** Rechecked thoroughly.

---

## Author Response (AR2)

**Response to Editor:** We are grateful to the editor for their diligent review of this article once again.

Our Reply follows (*the editor's comments are in italics and blue*)

*General Comments*

*Thank you for doing good work in reviewing the manuscript. There are a few minor corrections I still ask you to do.*

*Minor comments*

*One of the reviewers was asking about the soil NOx emissions and asked to add discussion on this. I understand this is difficult to take into account but you should add this as one of the discussion points as a source of uncertainty as asked by the reviewer. At the same tie time you could mention on seasonality of urban/rural factor as you find none*

**Reply:** Thanks for the comments. We will add discussion on these two points in the **Section 2.3**. Remarks are now added in page 7, line 29-33 and page 8, line 1-3 by saying: ***"The seasonality of Urban/Rural factors for NO₂ is also explored in Supplement Figure S3. It reveals that the Urban/Rural factor tends to be larger in spring and summer compared to autumn and winter, which contradicts the expected NO₂ lifetime. However, the difference is not significant and varies with changes in the research area. This could be attributed to the combined effects of factors like meteorological conditions, regional hotspots, human activities, biological sources, and topography. In addition, soil NOₓ emissions during summer can have a significant impact, particularly as they are a primary source for rural areas (Lu et al., 2021). However, it is challenging to provide concrete evidence based on the available data because we cannot distinguish the sources of NOₓ. Therefore, further refinement of the research area and consideration of multiple factors are necessary rather than concluding solely from ground observations."***

[Figure]

***Figure S3. The seasonal Urban/rural factors of ground NO₂. Each season is calculated from the average of 2015-2017 data. The blue dots correspond to the national scale, while the orange and green dots represent the NCP and YRD regions, respectively.***

*Another reviewer asked for language check and I do not see from the track changes that such correction would have been made. If you forgot to add those as track changes, please prove track changes version with those included.*

**Reply:** We thoroughly reviewed the language but did not highlight the minor changes. Now, we have provided a version with all track changes highlighted.

*Similarly updates on the reference links are not visible so it remains unclear which of those you have corrected and which ones not. Please include fully track-changed version of the manuscript.*

**Reply:** Sorry, we forgot to highlight the previous changes. The new version will include fully track changes.

*Minor additional comments*

*Equation 1 should have dot at the end*

**Reply:** Corrected.

*Equation 2 should have comma in the end*

**Reply:** Corrected.